# Gating and selectivity mechanisms for the lysosomal K$^+$ channel TMEM175

**SeCheol Oh, Navid Paknejad, Richard K Hite***

Structural Biology Program, Memorial Sloan Kettering Cancer Center, New York, United States

**Abstract** Transmembrane protein 175 (TMEM175) is a K$^+$-selective ion channel expressed in lysosomal membranes, where it establishes a membrane potential essential for lysosomal function and its dysregulation is associated with the development of Parkinson's Disease. TMEM175 is evolutionarily distinct from all known channels, predicting novel ion-selectivity and gating mechanisms. Here we present cryo-EM structures of human TMEM175 in open and closed conformations, enabled by resolutions up to 2.6 Å. Human TMEM175 adopts a homodimeric architecture with a central ion-conduction pore lined by the side chains of the pore-lining helices. Conserved isoleucine residues in the center of the pore serve as the gate in the closed conformation. In the widened channel in the open conformation, these same residues establish a constriction essential for K$^+$ selectivity. These studies reveal the mechanisms of permeation, selectivity and gating and lay the groundwork for understanding the role of TMEM175 in lysosomal function.

## Introduction

Lysosomes are small, acidic organelles that play essential roles in nutrient sensing, signaling, autophagy and degradation of macromolecules (*Ballabio and Bonifacino, 2020*; *Condon and Sabatini, 2019*; *Lawrence and Zoncu, 2019*; *Platt et al., 2018*). Many of these processes are intricately linked to ion transport across the membrane via numerous channels and transporters and defects in lysosomal transport proteins lead to a variety of diseases (*Grimm et al., 2017*; *Li et al., 2019*; *van Veen et al., 2020*). Transmembrane protein 175 (TMEM175) was recently identified as a constitutively-active potassium (K$^+$) selective channel expressed in lysosomal membranes responsible for establishing a membrane potential across the lysosomal membrane (*Cang et al., 2015*). TMEM175 is implicated in cellular proteostasis and its mutation is associated with the development of Parkinson's disease through a yet unknown mechanism, underscoring the importance of this channel to regulation of cellular homeostasis (*Blauwendraat et al., 2019*; *Cang et al., 2015*; *Iwaki et al., 2019*; *Jinn et al., 2019*; *Jinn et al., 2017*; *Krohn et al., 2020*; *Nalls et al., 2014*).

TMEM175 is evolutionarily distinct from known K$^+$ channels, displaying a unique membrane topology as well as lacking the conserved TVGYG selectivity filter present in canonical K$^+$ channels such as shaker K$^+$, BK and KcsA (*Doyle, 1998*; *Long et al., 2005*; *Tao et al., 2017*). In accordance with its divergent sequence, TMEM175 differs from canonical K$^+$ channels in its ion-permeation properties and its pharmaceutical sensitivities. Canonical K$^+$ channels are strongly selective for K$^+$ over Na$^+$, but are blocked by Cs$^+$. In contrast, TMEM175 permeates cations according to a lyotropic sequence, with Ca$^{2+}$ being least permeable, followed by Na$^+$, K$^+$ and Cs$^+$ being most permeable (*Cang et al., 2015*). Only one of the common K$^+$ channel inhibitors, 4-aminopyridine, can inhibit TMEM175 activity, while others, such as tetraethylammonium, do not alter channel activity (*Cang et al., 2015*). Together, these features predict unique ion permeation and selectivity mechanisms for TMEM175.

Structures of TMEM175 homologs from the prokaryotes *Chamaesiphon minutus* (*Lee et al., 2017*) and *Marivirga tractuosa* (*Brunner et al., 2018*) revealed that prokaryotic TMEM175 channels

***For correspondence:**
hiter@mskcc.org

**Competing interests:** The authors declare that no competing interests exist.

adopt a homotetrameric architecture with each protomer comprising a single 6-TM domain surrounding a central ion-conduction pathway. However, differences in the structures of the ion-conduction pathways led to differing proposals for how TMEM175 channels discriminate between cations, and thus ion selectivity in prokaryotic TMEM175 remains an open question. Moreover, unlike mammalian TMEM175 channels, prokaryotic TMEM175 channels are only minimally selective for $K^+$, permeating only 2 to 4 $K^+$ ions for every $Na^+$ ion compared to 36 for human TMEM175 (*Brunner et al., 2018*; *Cang et al., 2015*). Due to the differences in ion selectivity, it remains unknown whether the mechanisms proposed to govern prokaryotic TMEM175 channels are relevant to $K^+$ selectivity in mammalian TMEM175 channels. To elucidate the mechanisms underlying TMEM175 function in mammalian cells, we determined single-particle cryo-electron microscopic (cryo-EM) structures and analyzed the ion-permeation and selectivity properties of human TMEM175 (hTMEM175).

## Results

### Human TMEM175 is highly selective for $K^+$

To measure the ion selectivity of recombinant hTMEM175 channels, we took advantage of the observation that while hTMEM175 is endogenously expressed in the membranes of endosomes and lysosomes, transient overexpression as a GFP-fusion protein in HEK293T cells leads to expression of hTMEM175 at the plasma membrane (*Figure 1—figure supplement 1*; *Lee et al., 2017*). The permeation properties of these plasma membrane-localized hTMEM175 channels can be analyzed using whole-cell patch clamp. In a bi-ionic condition, in which the pipette (intracellular) solution contains 150 mM $K^+$ and the bath (extracellular) solution contains 150 mM $Na^+$, hTMEM175 displays a strong preference for $K^+$ (*Figure 1A*). The reversal potential calculated from voltage families stepping from −100 mV to +100 mV was −55 ± 2.7 mV, corresponding to a $K^+$/$Na^+$ permeation ratio ($P_K/P_{Na}$) of ~9 (*Figure 1B*). Consistent with previous results (*Cang et al., 2015*), hTMEM175 is also selective for $Cs^+$ over $Na^+$. In a $Cs^+$/$Na^+$ bi-ionic condition, the mean reversal potential of hTMEM175 is −65 ± 4.1 mV ($P_{Cs}/P_{Na}$ of ~13) (*Figure 1C–D*). We also measured whole-cell currents from non-transfected HEK293T cells, which revealed the presence of non-selective currents whose magnitude varied between 50 and 100 pA at +100 mV (*Figure 1—figure supplement 1*). Because these endogenous currents are also present in the hTMEM175 transfected cells, the ion-selectivity measurements determined using whole-cell patch clamp underrepresent the selectivity of hTMEM175 and the true values are likely closer to those measured in endolysosomal patch clamp ($P_K/P_{Na}$ ~36) (*Cang et al., 2015*).

We next overexpressed hTMEM175 in HEK293S GnTi- cells and purified it to homogeneity (*Figure 2—figure supplement 1*). To assess the activity of the purified channels, we reconstituted hTMEM175 into proteoliposomes composed of a 3:1 ratio of 1-palmitoyl-2-oleoyl-sn-glycero-3-phosphoethanolamine (POPE) and 1-palmitoyl-2-oleoyl-sn-glycero-3-phospho-(1'-rac-glycerol) (POPG) and measured channel activity. Using a 9-amino-6-chloro-2-methoxyacridine (ACMA)-based flux assay (*Su et al., 2016*) with 300 mM $K^+$ inside of the vesicles and 300 mM $Na^+$ outside, robust $K^+$ efflux was detected from proteoliposomes containing hTMEM175 following the addition of the ionophore carbonyl cyanide m-chlorophenylhydrazone (CCCP) compared to empty liposomes (*Figure 1E*). No flux could be detected when the inhibitor 4-aminopyridine (4-AP) was added to the proteoliposomes at a concentration of 1 mM, demonstrating that reconstituted hTMEM175 channels are active and that they retain their $K^+$ selectivity and their pharmacological sensitivity to 4-AP.

### Structure of hTMEM175

To investigate the mechanisms that govern hTMEM175 function, we collected cryo-EM images of hTMEM175 purified in 150 mM $K^+$. Three-dimensional classification revealed that two conformations were present among the imaged particles; class 1, which was resolved at a resolution of 2.6 Å, and class 2, which was resolved at a resolution of 3.0 Å (*Figure 2A* and *Figure 2—figure supplement 1*, *2* and *Table 1*). Due to the high degree of similarity between the two classes, we will first describe the higher-resolution class 1 structure (*Figure 2B*). hTMEM175 is composed of two homologous 6-helix repeat domains that share ~23% sequence identity (*Figure 2C* and *Figure 2—figure supplement 3*). Consequently, the density map revealed that while hTMEM175 is homodimeric, similarities between the two 6-helix repeat domains result in a pseudo-four-fold symmetric architecture.

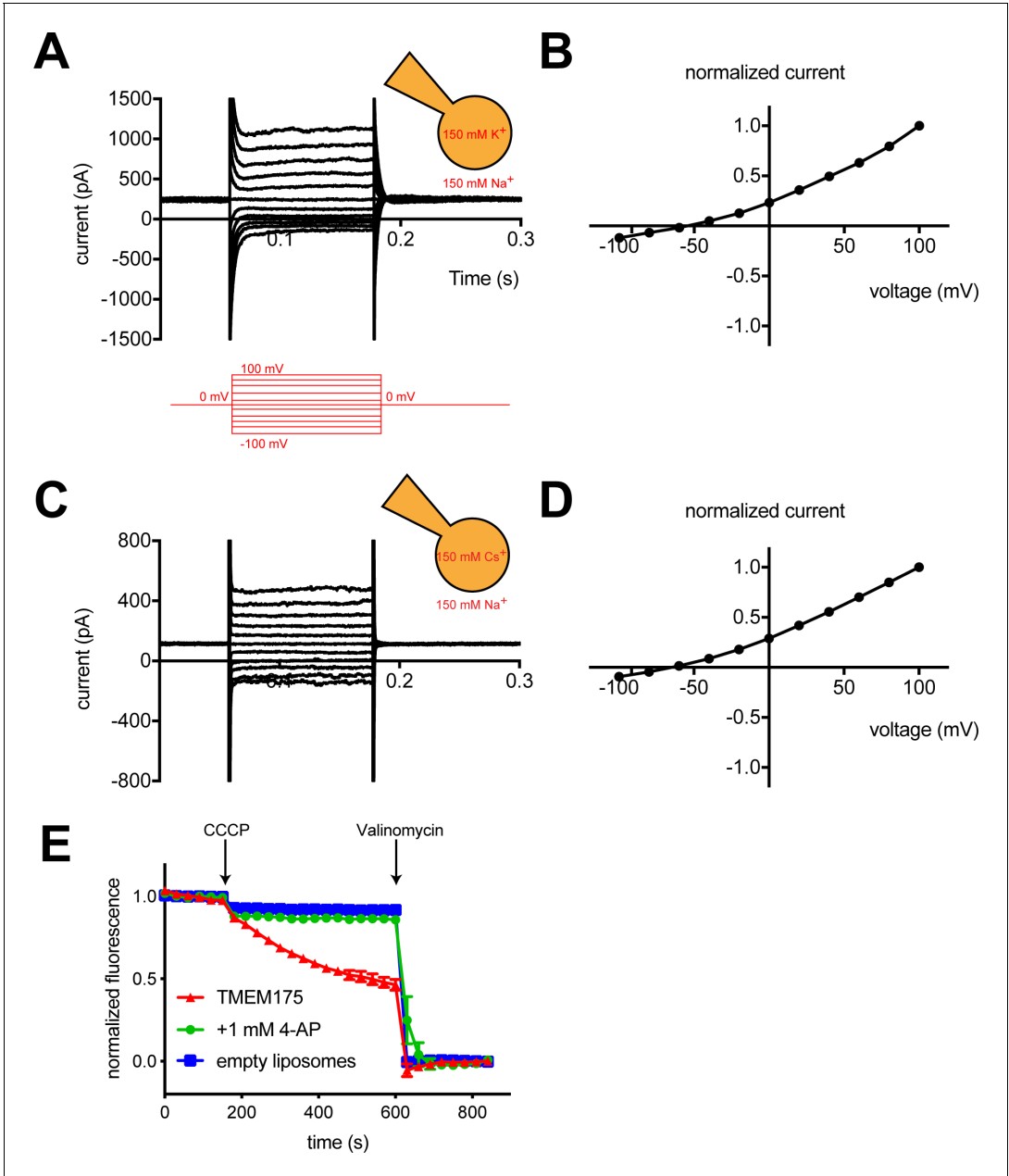

**Figure 1.** hTMEM175 is a K$^+$ selective channel. (**A, C**) Representative whole-cell electrical recordings of hTMEM175-transfected HEK293T cells. In bi-ionic conditions of 150 mM K$^+$ (intracellular) and 150 mM Na$^+$ (extracellular) (**A**) or 150 mM Cs$^+$ (intracellular) and 150 mM Na$^+$ (extracellular) (**C**), currents were measured using the following protocol (red): from a holding potential of 0 mV, the voltage was stepped to voltages between −100 and +100 mV, in 20 mV increments, then returned to 0 mV. (**B, D**) Normalized current-voltage relationships of three independent whole-cell patch clamp recordings of hTMEM175-transfected HEK293T cells in bi-ionic conditions of 150 mM K$^+$ (intracellular) and 150 mM Na$^+$ (extracellular) (**B**) or 150 mM Cs$^+$ (intracellular) and 150 mM Na$^+$ (extracellular) (**D**). (**E**) K$^+$ efflux from purified hTMEM175 reconstituted into liposomes in the presence or absence of 1 mM 4-aminopyridine and from empty liposomes was monitored using a fluorescence-based flux assay. Arrows mark addition of the proton ionophore CCCP to initiate K$^+$ flux and addition of the K$^+$ ionophore valinomycin to measure total flux capacity of the liposomes. All experiments were performed in triplicate and error bars represent SEM.

The online version of this article includes the following figure supplement(s) for figure 1:

**Figure supplement 1.** Electrophysiological analysis of hTMEM175 in HEK293T cells.

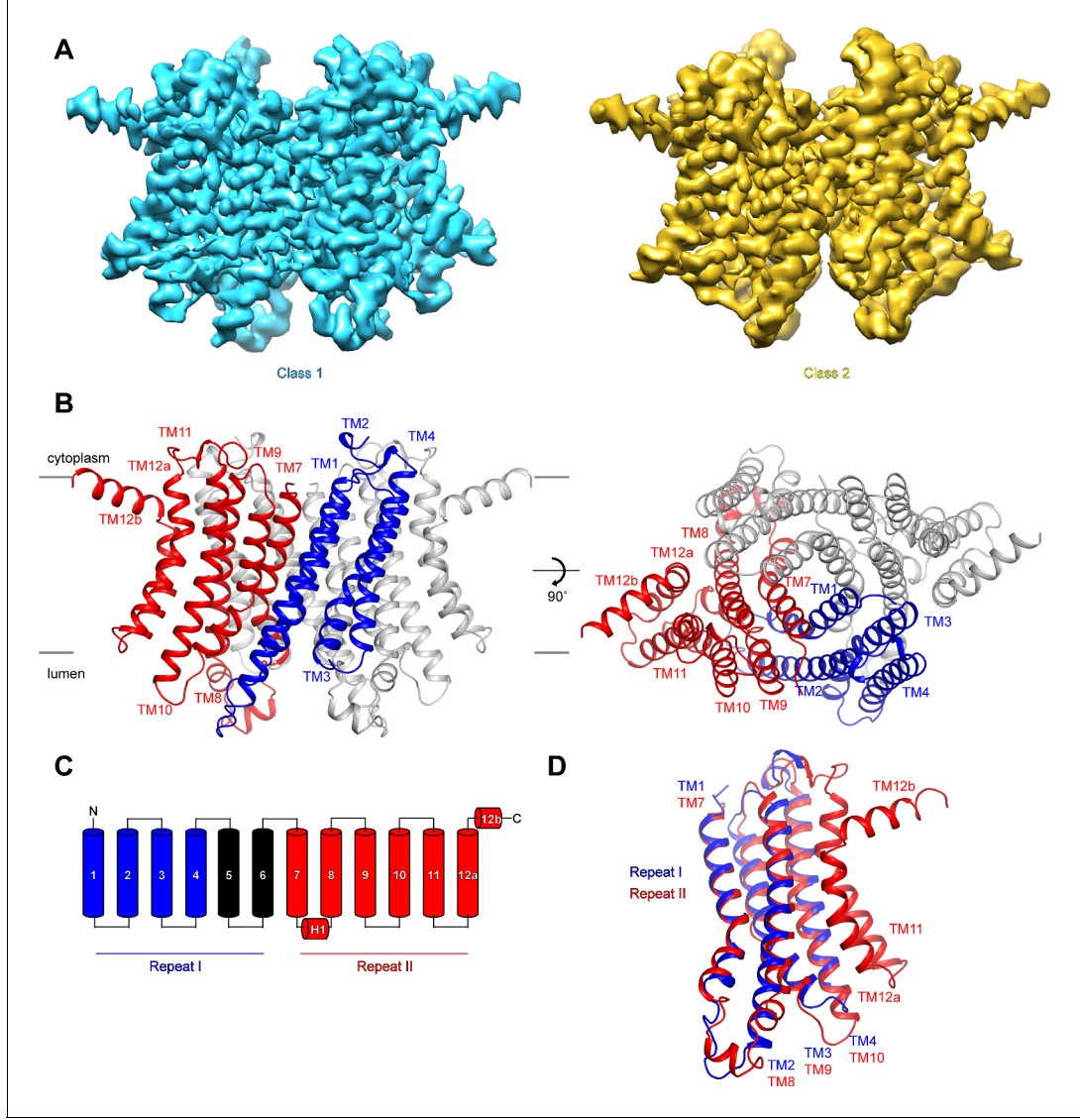

**Figure 2.** Structure of hTMEM175. (**A**) Cryo-EM density maps of class 1 (cyan) and class 2 (gold) hTMEM175 in KCl depicted from within the membrane. (**B**) Structure of class 1 hTMEM175 depicted from within the membrane (left) and from the cytoplasm (right). TM1-TM4 (repeat I) and TM7-TM12 (repeat II) of protomer A are shown in blue and red, respectively. Protomer B is shown in grey. Approximate width of the lipid bilayer is shown as grey bars. (**C**) Topology of hTMEM175. TM1-TM4 (repeat I) and TM7-TM12 (repeat II) are shown in blue and red, respectively. Unmodelled helices TM5 and TM6 are shown in black. (**D**) Superposition of class 1 hTMEM175 repeat I (blue) with repeat II (red).

The online version of this article includes the following figure supplement(s) for figure 2:

**Figure supplement 1.** Cryo-EM analysis of hTMEM175 in KCl.

**Figure supplement 2.** Cryo-EM densities in hTMEM175.

**Figure supplement 3.** Sequence alignment of TMEM175 channels.

**Figure supplement 4.** TMEM175 channels share a common fold.

Inspection of the map identified features that were sufficiently well resolved to permit us to distinguish repeat I (TM1-TM6) from repeat II (TM7-TM12) and build a de novo structure of hTMEM175, comprising TM1-TM4 and TM7-TM12. The densities corresponding to peripheral helices TM5 and TM6 are too poorly resolved in the map for modelling (*Figure 2—figure supplement 2*). Viewed from the cytoplasm, hTMEM175 is a diamond-shaped channel measuring ~85 Å along the long axis and ~60 Å along the short axis with the ion-conduction pathway located at the center of the channel along the pseudo-four-fold axis (*Figure 2B*). The six helices of each repeat form distinct domains and no swapping of domains or helices is evident. Viewed from within the plane of the membrane,

**Table 1.** Cryo-EM data acquisition, reconstruction and model refinement statistics.

| | hTMEM175 Class 1 K$^+$ | hTMEM175 Class 2 K$^+$ | hTMEM175 Class 1 Cs$^+$ | hTMEM175 Class 2 Cs$^+$ |
|---|---|---|---|---|
| **Cryo-EM acquisition and processing** | | | | |
| EMDB accession # | 21603 | 21604 | 21605 | 21606 |
| Magnification | 22,500x | 22,500x | 22,500x | 22,500x |
| Voltage (kV) | 300 | 300 | 300 | 300 |
| Total electron exposure (e$^-$ / Å$^2$) | 61 | 61 | 61 | 61 |
| Exposure time (s) | 8 | 8 | 8 | 8 |
| Defocus range (uM) | -1.0 to -2.5 | -1.0 to -2.5 | -1.0 to -2.5 | -1.0 to -2.5 |
| Pixel size (Å) | 1.088 | 1.088 | 1.088 | 1.088 |
| Symmetry imposed | C2 | C2 | C2 | C2 |
| Initial particles | 4,153,614 | 4,153,614 | 4,275,219 | 4,275,219 |
| Final particles | 342,340 | 57,152 | 94,653 | 70,132 |
| Resolution (masked FSC = 0.143, Å) | 2.64 | 3.03 | 3.17 | 3.24 |
| Density modified CC (0.5, Å) | 2.67 | 3.09 | 3.12 | 3.23 |
| **Model Refinement** | | | | |
| PDB ID | 6WC9 | 6WCA | 6WCB | 6WCC |
| Model resolution (FSC = 0.50/0.143Å) | 2.68 / 2.32 | 3.07 / 2.67 | 3.18 / 2.71 | 3.27 / 2.84 |
| Model refinement resolution | 300-2.6 | 300-3.0 | 300-3.2 | 300-3.2 |
| RMS deviations | | | | |
|  Bond length (Å) | 0.005 | 0.002 | 0.004 | 0.003 |
|  Bond angle (°) | 0.532 | 0.507 | 0.406 | 0.506 |
| Ramachandran plot | | | | |
|  Favored (%) | 96.13 | 96.42 | 99.17 | 99.17 |
|  Allowed (%) | 3.87 | 3.58 | 0.83 | 0.83 |
|  Disallowed (%) | 0 | 0 | 0 | 0 |
| Rotamer Outliers (%) | 2.27 | 1.61 | 1.29 | 2.26 |
| Validation | | | | |
|  MolProbity score | 1.71 | 1.74 | 1.12 | 1.49 |
|  Clashscore | 3.88 | 6.28 | 2.50 | 4.39 |

most of hTMEM175 is embedded within the membrane with only short loops extending out of the membrane on either side (*Figure 2B*).

Consistent with the high sequence homology between repeat I (TM1-TM6) and repeat II (TM7-TM12), alignment reveals that their structures are nearly identical with an RMSD of 1.7 Å (*Figure 2D*). The structures of repeat I and repeat II are also homologous with the monomeric structures of TMEM175 channels from the prokaryotes *Chamaesiphon minutus* (TMEM175$_{Cm}$) (*Lee et al., 2017*) and *Marivirga tractuosa* (TMEM175$_{Mt}$) (*Brunner et al., 2018*; *Figure 2—figure supplement 4*). Thus, while prokaryotic TMEM175 channels are homotetramers rather than homodimers, the global architecture of TMEM175 channels is conserved.

## Ion-conduction pathway contains ordered ions and waters

The ion-conduction pathway of hTMEM175 is located along the central axis of the channel extending approximately 45 Å from the cytoplasm to the lysosomal lumen (*Figure 3A*) The pore is lined by the side chains of the kinked pore-lining helices, TM1 and TM7. The side chains of TM1 and TM7 create multiple constrictions whose radii are less than 2.0 Å and would restrict the permeation of hydrated K$^+$ ions. The narrowest of these constrictions is formed by the side chains of Ile46 from TM1 and

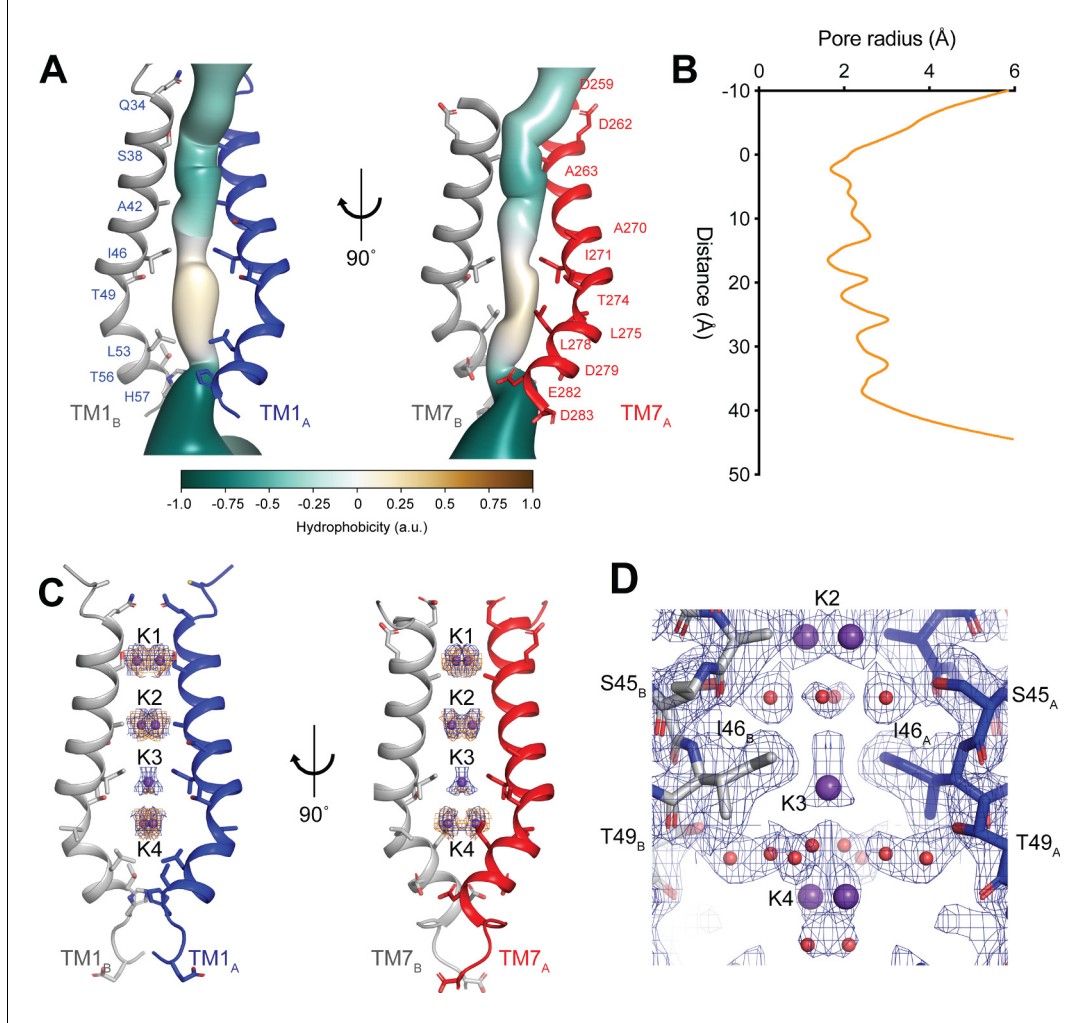

**Figure 3.** hTMEM175 ion conduction pathway. (**A**) Ion permeation pathway of class 1 hTMEM175. Pore-lining helices TM1 from protomers A (blue) and B (grey) are shown at left and TM7 from protomers A (red) and B (grey) are shown at right with all other helices removed for clarity. Pore-lining residues are shown as sticks. Surface representation of the ion permeation pathway colored by hydrophobicity calculated using the class 1 structure without ions and water molecules calculated using CHAP (*Klesse et al., 2019*). (**B**) Dimensions of the ion conduction pathway in class 1 calculated using CHAP (*Klesse et al., 2019*). (**C**) Overlapping non-protein density peaks in the ion permeation pathway of class 1 in the presence of K$^+$ (blue mesh, 12 σ threshold) and Cs$^+$ (gold mesh, 8 σ threshold). hTMEM175 is shown as in **A**. K$^+$ ions are shown as violet spheres. (**D**) Density map near the isoleucine constriction displayed as blue mesh and contoured at 12 σ threshold. K$^+$ ions are shown as violet spheres and water molecules are shown as red spheres.

The online version of this article includes the following figure supplement(s) for figure 3:

**Figure supplement 1.** Non-protein densities in hTMEM175 density maps.
**Figure supplement 2.** Cryo-EM analysis of hTMEM175 in CsCl.

Ile271 from TM7, which are conserved among all eukaryotic TMEM175 channels, and has minimum radius of 1.7 Å (*Figure 3B*). Just below the constriction formed by Ile46 and Ile271, the pore contains an expanded vestibule that is also lined by the side chains of Thr49 and Leu53 from TM1 and Thr274, Leu275 and Leu278, making it much more hydrophobic than the rest of the pore (*Figure 3A*). Despite its hydrophobicity, multiple non-protein density peaks are resolved within the vestibule (*Figure 3—figure supplement 1*). Non-protein density peaks are also present in the vestibule in a density map calculated without symmetry, indicating that they represent ordered molecules rather than arising from the accumulation of noise along the two-fold symmetry axis during image processing (*Figure 3—figure supplement 1*).

In addition to the non-protein densities resolved in the hydrophobic vestibule, numerous other non-protein densities are resolved in the other, more hydrophilic regions of the pore (*Figure 3—figure supplement 1*). However, due to the large number of non-protein peaks and lack of obvious protein-coordinated ion-binding sites, it was not possible to unambiguously distinguish ions from water molecules based on the density map alone. To aid in assigning the identity of these peaks, we collected cryo-EM images of hTMEM175 purified in 150 mM $Cs^+$. We chose to determine structures in the presence of $Cs^+$ for two reasons. First, $Cs^+$ scatters electrons approximately three times more strongly than $K^+$ (*Peng, 1998*) and thus bound $Cs^+$ ions should yield density peaks that can be distinguished from those corresponding to water and other non-protein atoms in the density map. Second, because the permeation of $Cs^+$ is similar to $K^+$, we hypothesized that $Cs^+$ would occupy the same binding sites in the pore as does $K^+$ and thus facilitate identification of the ion-binding sites (*Figure 1*).

Three-dimensional classification revealed that hTMEM175 adopts the same two conformations in the presence of $Cs^+$ that exist in the presence of $K^+$ (*Figure 3—figure supplement 2* and *Table 1*). A number of non-protein densities are resolved in the pore of class 1 hTMEM175 in the presence of $Cs^+$. Because of the similarities between the class 1 structures determined in $Cs^+$ and $K^+$ (all-atom RMSD = 0.2 Å), we could directly compare the density maps at a threshold of σ = 12 for the class 1 map determined in $K^+$ and of σ = 8 for the class 1 map determined in $Cs^+$, identifying four overlapping non-protein density peaks in both maps (*Figure 3C*). We therefore modelled these overlapping densities as ions, either $K^+$ or $Cs^+$ depending on the condition. We assigned the remaining non-protein densities as ordered waters. Despite the narrow dimensions of the pore, the protein only minimally participates in the direct coordination of the bound ions (*Figure 3D*). Instead, ordered water molecules form nearly all of the direct interactions with the bound ions. Near the cytoplasmic entrance to the pore, the ion in the K1 site is directly coordinated by the side chain of Ser38 and indirectly coordinated by the side chain of Asp266 and the backbone oxygens of Ser38 and Glu259 via ordered waters (*Figure 4A*). The ion in the K2 site is indirectly coordinated by the side chain of Ser45 and the backbone oxygens of Ala263 and Gly267 via ordered waters (*Figure 4B*). The K3 site, the weakest of the four sites in both density maps, is located near the isoleucine constriction that is flanked by two layers of four water molecules (*Figure 4C*). The waters on the cytoplasmic side are coordinated by the side chain of Ser45 and the backbone oxygen of Gly267, while the waters on the luminal side are indirectly coordinated by the side chains of Thr49 and Thr274 and by the backbone oxygens of Ile46 and Ile271. Unlike the spherical densities resolved at the other ion-binding sites, the density for the K3 ion is elongated and extends between the two layers of waters on either side of the isoleucine constriction (*Figure 3D*). The waters on the luminal side of the constriction are located between 3.1 and 3.4 Å away from the center of the luminal portion of the density peak, while the waters on the cytoplasmic side are located between 3.1 and 3.2 Å away from the center of the cytoplasmic portion of the density peak. Because the center of the constriction is hydrophobic, ions are unlikely to stably bind there, and so we speculate that the K3 density corresponds to the sum of two partially-occupied ion-binding sites positioned on either the cytoplasmic or the luminal side of the constriction where the ions can be directly coordinated by the nearby waters. The fourth ion-binding site (K4) is located in the hydrophobic vestibule near the luminal entrance to the pore (*Figure 4D*). Notably, two symmetry-related copies of sites K1, K2 and K4 are present in the structure because the K1, K2 and K4 sites are located slightly off of the central axis of the pore (*Figure 4*). However, the distances between the pairs of K1, K2 and K4 ion-binding sites are 3.0 Å, 2.9 Å and 3.3 Å, respectively, which are likely too close for both symmetry-related sites to be simultaneously occupied with ions.

## Structural heterogeneity reveals gating mechanism

To better understand the functional states of the two conformations resolved in our data sets, we next superimposed the class 1 and class 2 structures determined in the presence of $K^+$ (*Figure 5A*). Overall, the two classes are very similar, with an all-atom RMSD of 0.9 Å. The similarities are especially pronounced in the cytoplasmic side of the channel, which likely arises from the existence of interaction networks at the intra-subunit interfaces between repeats I and II and at the inter-subunit interfaces between protomers (*Figure 5B*). These interaction networks adopt identical configurations in both conformations and are anchored by the essential RxxxFSD motif on TM1 and TM7 (*Cang et al., 2015*). In addition to the RxxxFSD motif, the networks involve a conserved histidine

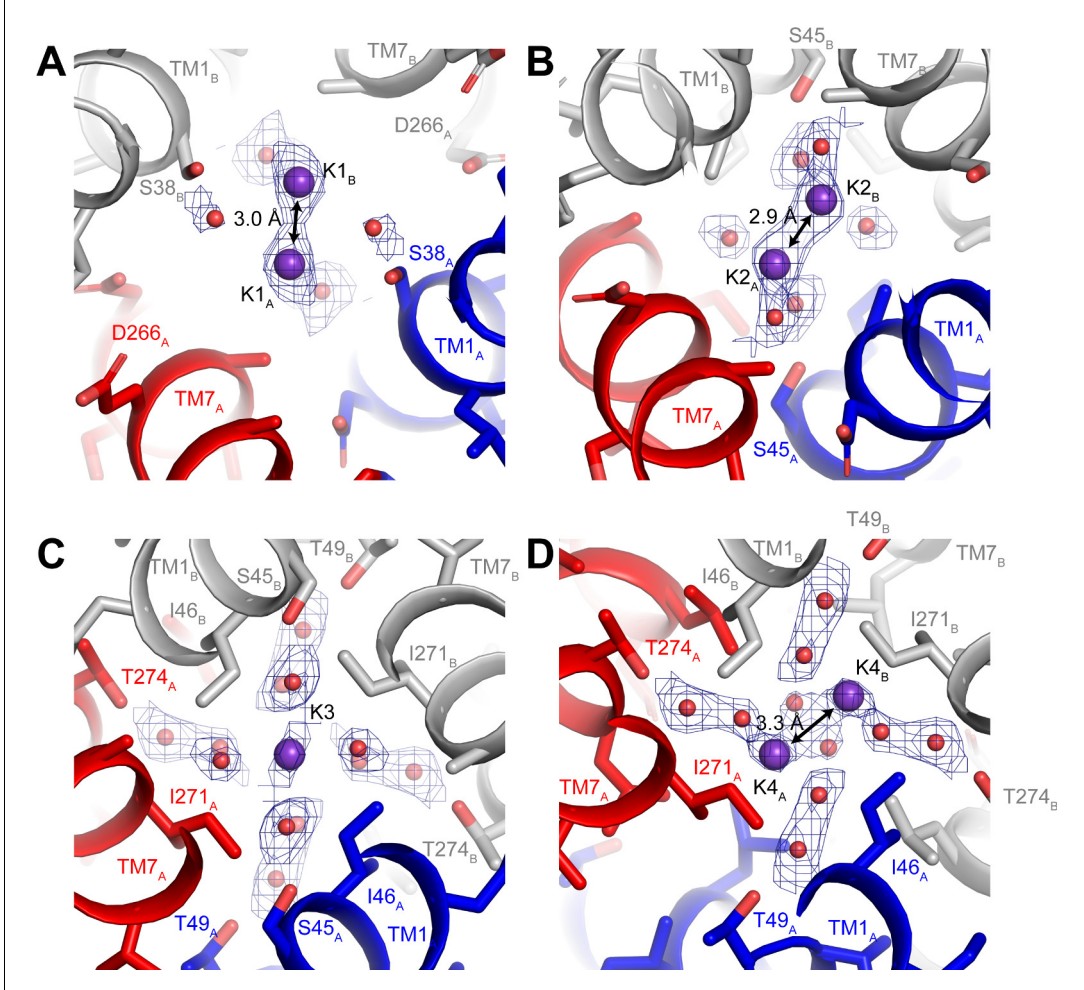

**Figure 4.** Ion-binding sites in hTMEM175. Structure of the K1 (**A**), K2 (**B**), K3 (**C**) and K4 (**D**) binding sites in class 1. K$^+$ ions are shown as violet spheres and water molecules are shown as red spheres. Density for K$^+$ and water molecules shown as blue mesh and contoured at 12 σ threshold.

and a conserved tryptophan on TM2/TM8 and a conserved asparagine on TM3/TM9 (*Figure 2—figure supplement 3*). Similar interaction networks were resolved in the prokaryotic structures (*Brunner et al., 2018*; *Lee et al., 2017*), suggesting a conserved role for the RxxxFSD motif in maintaining channel quaternary structure during conformational changes.

In contrast to the rigid cytoplasmic side, differences between the two classes can be readily detected on the luminal side of the channel (*Figure 5A* and *Video 1*). When viewed from the luminal side of the channel, the ends of the transmembrane helices in the class 2 structure are rotated in a clockwise manner compared to their positions in class 1 (*Figure 5A*). The luminal loops between the transmembrane helices also adopt different conformations, with the loop between TM9 and TM10 undergoing the largest change. In class 2, the last turn of TM9 is unwound and moves nearly 12 Å from its position adjacent to the loop between TM11 and TM12 in class 1 to interact with the loop between TM1 and TM2.

Inspection of the pore-lining helices, TM1 and TM7, reveals that the clockwise rotation of their luminal ends from class 1 to class 2 is accompanied by the adoption of a straighter, α-helical conformation, particularly for TM7 (*Figure 5C*). In class 1, the kink in TM1 is stabilized by Pro54 and by a water molecule coordinated by the side chain of Thr84 on TM2, the backbone carbonyl oxygen of Ala48 and the backbone amide nitrogen of Met51, while the kink in TM7 is stabilized by a water molecule coordinated by the side chain of Ser316 on TM8, the backbone carbonyl oxygens of Val272 and Ala273 and the backbone amide nitrogen of Leu276 (*Figure 5—figure supplement 1*). In class 2, the waters stabilizing the kinks are displaced by the side chains of Met51 and Leu276. The

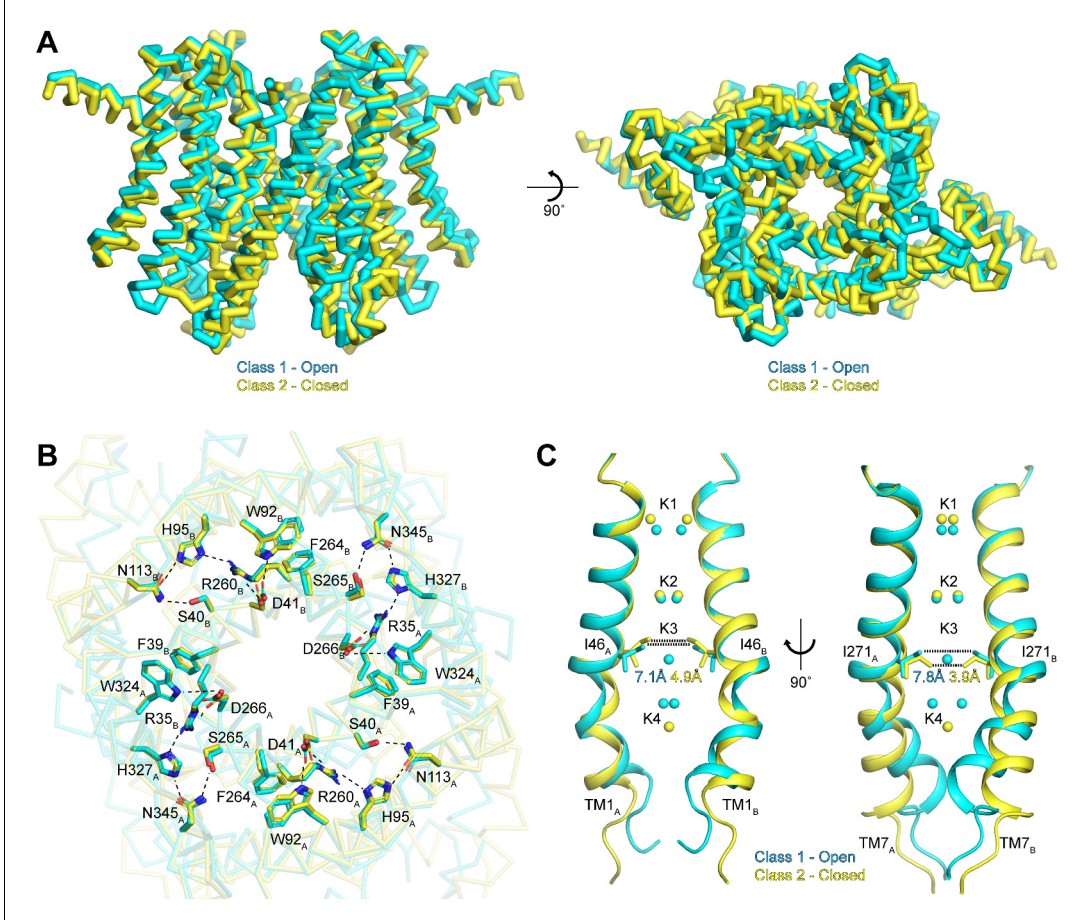

**Figure 5.** Gating in hTMEM175. (**A**) Superposition of class 1 (cyan) and class 2 (gold) viewed from within the membrane (left) and from the lysosomal lumen (right). (**B**) Alignment of the RxxxFSD inter- and intra-subunit interaction networks in class 1 (cyan) and class 2 (gold) depicted as sticks and viewed from the cytosol. Ionic and polar interaction are shown as dashed lines. (**C**) Ion conduction pathways of class 1 (cyan) and class 2 (gold). TM1 is shown at left and TM7 is shown at right with all other helices removed for clarity. K$^+$ ion binding sites are shown as spheres. Dotted lines correspond to minimum distance between opposing residues at the isoleucine constriction.

The online version of this article includes the following figure supplement(s) for figure 5:

**Figure supplement 1.** Water molecules stabilize kinked conformation of TM1 and TM7 in class 1.

resultant straightening of TM1 and TM7 in class 2 alters the shape of the pore, particularly at the isoleucine constriction (*Figure 5C*). In class 1, the minimum radius of the isoleucine constriction is 1.7 Å with an ion-binding site in the center surrounded by the isoleucine side chains in a nearly four-fold symmetric configuration. In class 2, the four-fold arrangement of the isoleucine side chains is broken by an inward movement of all four isoleucine residues and a rotation of the Ile271 side chains. These changes reduce the minimum pore radius to 0.5 Å, which is too narrow to accommodate dehydrated K$^+$ ions. Accordingly, no density is resolved in the K3 ion-binding site in class 2. Thus, class 2 represents a closed conformation with Ile46 and Ile271 forming the channel gate.

If class 2 represents a closed conformation, what state does class 1 represent? The proteoliposome flux assay demonstrated that purified hTMEM175 can conduct ions in the absence of stimuli, indicating that hTMEM175 can adopt an open state in the similar conditions used for cryo-EM analysis (*Figure 1E*). However, in order for class 1 to be a conductive state, ions would have to be able to permeate its narrow pore in a partially dehydrated state. In the cytosolic and luminal regions of the pore, the bound ions are coordinated by numerous water molecules and the constrictions are formed by polar and charged side chains, suggesting that partially hydrated ions can readily translocate. In contrast, ions would have to be almost completely dehydrated to penetrate the isoleucine constriction because of its size and hydrophobicity. However, the dehydration need only be transient

Open

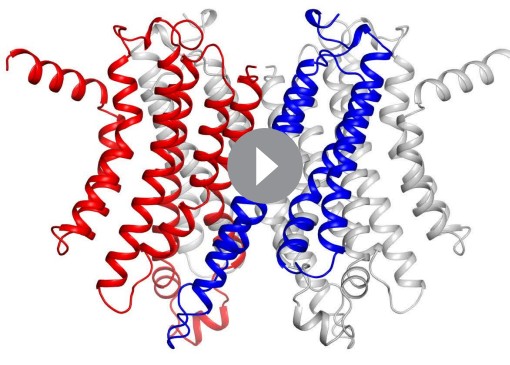

**Video 1.** Morph between open class 1 and closed class 2. https://elifesciences.org/articles/53430#video1

due to the layers of water molecules on either side of the isoleucine constriction that can rehydrate the ion once it passes through the constriction (*Figure 6A*).

To determine if the ordered water molecules facilitate ion permeation through the isoleucine constriction, we mutated Ser45 or Thr274, conserved residues on the cytosolic side and luminal side of the isoleucine constriction, respectively, whose side-chain hydroxyl groups participate in the coordination of ordered waters (*Figure 6A*). We first analyzed the effects of the mutations on protein stability using fluorescence size-exclusion chromatography (*Goehring et al., 2014*), which revealed that channels with mutations to Ser45 and Thr274 are properly folded as dimers and express at levels within two-fold of wild-type hTMEM175 (*Figure 6—figure supplement 1*). We next assessed the effects of the mutations on channel activity using whole-cell patch clamp in a bi-ionic $Cs^+/Na^+$ condition. Currents recorded from cells expressing the S45A and T274V mutants, which lack one of the hydroxyl groups involved in water coordination, were

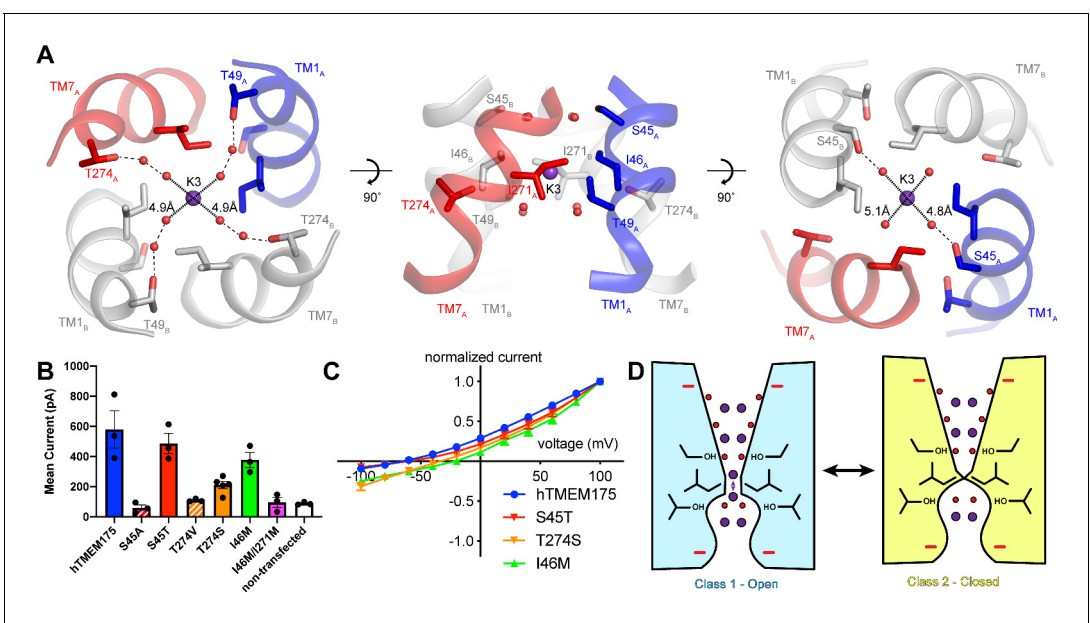

**Figure 6.** Permeation and selectivity through the isoleucine constriction. (**A**) The isoleucine constriction is flanked by two layers of ordered water molecules. The cytosolic layer of waters is partially coordinated by Ser45, while the luminal layer is partially coordinated by Thr49 and Thr274. (**B**) Mean current recorded from HEK293T cells transfected with hTMEM175 (blue), S45A (red dashed), S45T (red), T274V (orange dashed), T274S (orange), I46M (green), I46M/I271M (magenta) and non-transfected (white) at +100 mV in a bi-ionic condition of 150 mM $Cs^+$ (intracellular) and 150 mM $Na^+$ (extracellular). (**C**) Normalized I-V relationship of whole-cell patch clamp of hTMEM175 transfected (blue), S45T transfected (red), T274S transfected (orange) and I46M transfected (green) HEK293T cells in a bi-ionic condition of 150 mM $Cs^+$ (intracellular) and 150 mM $Na^+$ (extracellular). All experiments were performed at least three times and error bars represent SEM. (**D**) Model for ion selectivity and gating in hTMEM175. In the open state, ions are transiently dehydrated through the isoleucine constriction, favoring permeation of $K^+$ ions. In the closed state, the isoleucine constriction closes, preventing ion permeation.

The online version of this article includes the following figure supplement(s) for figure 6:

**Figure supplement 1.** Functional analysis of hTMEM175 mutants.

indistinguishable from those recorded from non-transfected cells (*Figure 6B*). In contrast, cells expressing either the S45T or the T274S mutant yielded Cs$^+$-selective currents (*Figure 6B–C*). These results reveal a critical role for residues that coordinate water molecules in facilitating the permeation of ions through the isoleucine constriction and suggest that the water molecules themselves may participate in ion permeation. We therefore speculate that ions can permeate through the isoleucine constriction of class 1 in a partially hydrated state and that class 1 represents a conductive state.

## Mechanisms of ion selectivity

The transient dehydration of ions through the isoleucine constriction implies a mechanism for ion selectivity. Because the enthalpies of dehydration for Cs$^+$ (250 kJ/mol) and K$^+$ (295 kJ/mol) are lower than that of Na$^+$ (365 kJ/mol) (*Marcus, 1991*), Cs$^+$ and K$^+$ ions can more readily access the partially dehydrated state necessary to permeate through the isoleucine constriction and are thus permeated more efficiently than Na$^+$. Previously, mutation of Ile46 and Ile271 to asparagine was shown to diminish ion selectivity, which led to the proposal that the Ile46 and Ile271 act as a hydrophobic selectivity filter (*Lee et al., 2017*). However, in the class 1 structure of hTMEM175 the branched side chains of Ile46 and Ile271 form a constriction that we propose is precisely shaped to allow dehydrated ions to permeate, suggesting that the unique shape of isoleucine may also be essential (*Figure 3D*). To test if hydrophobicity itself is sufficient to impart ion selectivity, we mutated Ile46 to methionine, which is similar to isoleucine in terms of volume occupied and hydrophobicity, and recorded whole-cell currents. In a bi-ionic Cs$^+$/Na$^+$ condition, the I46M mutant displayed less selectivity for Cs$^+$ with a mean reversal potential of $-21 \pm 1.5$ mV compared to $-65 \pm 4.1$ mV for wild-type hTMEM175 (*Figure 6C*). We also attempted to record whole-cell currents from cells expressing the I46M/I271M double mutant. However, no exogenous currents could be detected, consistent with a previous report that mutations of Ile46 and Ile271 to alanine, valine, leucine and phenylalanine were not functional (*Lee et al., 2017*). These results point to a unique role for isoleucine side chains in establishing a selectivity filter that cannot be duplicated by other amino acids.

Although the isoleucine constriction participates in the selective permeation of ions by hTMEM175, it is unclear if the energetic differences between K$^+$ and Na$^+$ dehydration are alone sufficient to generate a nearly forty-fold selectivity for K$^+$ over Na$^+$. Moreover, alignment of hTMEM175 with prokaryotic homologs reveal that isoleucine residues are broadly conserved at the position of the isoleucine constrictions (*Brunner et al., 2018*; *Lee et al., 2017*; *Figure 2—figure supplement 3*), suggesting that the isoleucine constriction may be a conserved feature among TMEM175 channels. Given the greater selectivity of hTMEM175 compared to its prokaryotic homologs (*Brunner et al., 2018*; *Lee et al., 2017*), additional, unknown mechanisms may further contribute to K$^+$ selectivity in hTMEM175. Because the ion-binding sites are largely formed by waters, we hypothesized that ion coordination by waters might represent an additional mechanism of selectivity in hTMEM175. Comparing the structures of the water-mediated ion-binding sites with the structure of the ion-binding sites in KcsA revealed that the distances between water molecules in the layers on either side of the isoleucine constriction in hTMEM175 are 4.8 Å and 5.1 Å, similar to the distances between opposing backbone carbonyl oxygens that form the ion-binding sites in the TVGYG selectivity filter of KcsA (4.5–5.1 Å) (*Zhou et al., 2001*; *Figure 6A*). The similarity of these structures suggests that the water molecules in hTMEM175 may play an analogous role in hTMEM175 and participate in ion selectivity. To examine the influence of the ordered water molecules near the isoleucine constriction on ion selectivity, we compared the reversal potential of S45T and T274S mutants with wild-type hTMEM175. While the reversal potential of the S45T was similar to wild type ($-62 \pm 1.9$ mV), it was right-shifted to $-34 \pm 5.1$ mV for the T274S mutant, suggesting that the T274S mutant is less selective than wild-type hTMEM175 (*Figure 6B–C*). These results suggest that minor differences in residues that coordinate water molecules near the isoleucine constriction can influence ion selectivity. Because Thr274 does not directly interact with the permeating ions, these results suggest that the network of ordered molecules in the pore may also contribute to ion selectivity in hTMEM175 channels.

## Discussion

To gain insights into the mechanisms underlying TMEM175 channel function, we determined cryo-EM structures of hTMEM175 in two conformations. Numerous non-protein densities are resolved in the ion conduction pathways of both structures. However, unlike anomalous X-ray diffraction experiments, which can specifically localize atoms in protein structures, ligand identification in cryo-EM structures relies on examination of the local environment of the binding sites. While such analyses can be straightforward for well-characterized proteins, densities can be ambiguous in structures of less well-characterized proteins. Because TMEM175 channels are evolutionarily distinct from all other known ion channels, we needed to an alternative approach to distinguish between the ion densities in the pore and ordered waters. Therefore, we determined structures of hTMEM175 in the presence of two ions, $K^+$ and $Cs^+$, which are permeated similarly by hTMEM175, but have different electron scattering properties (*Figure 1*). The comparison revealed four overlapping peaks in the class 1 maps and three in the class 2 maps (*Figure 3*). While this approach enabled us to better identify ion densities and may be useful for identifying ion-binding sites in other proteins, the signal is much weaker than that generated by anomalous X-ray diffraction experiments and future efforts will be required to improve identification of ligands in cryo-EM density maps.

Identification of the ion-binding sites enabled us to tentatively assign the two conformations as an open state and a closed state. However, cryo-EM structures represent snapshots of the protein being imaged and it is difficult to explicitly assign functional states to the conformational states resolved. Because the population of imaged particles represents the equilibrium of states at the condition being imaged, it is sometimes possible to infer that the fraction of particles in a particular structural state corresponds to the likelihood of the corresponding functional state (*Hite and MacKinnon, 2017*). For small proteins like hTMEM175, such correspondence is difficult due to the large number of particles that are removed from the data sets during classification and thus whose conformational state is unknown (*Table 1*). Moreover, while care is taken during image processing to identify all conformations, it is possible that additional low-abundance conformations remain unidentified in the data sets. Thus, future orthogonal experiments will be necessary to test the functional assignments that we propose for our structures as well as the mechanisms derived from their interpretation.

In contrast to most ion channels, for which gating and selectivity are physically uncoupled, the precise three-dimensional arrangement of Ile46 and Ile271 in the two hTMEM175 structures suggests that they serve as both the gating residues as well as the ion selectivity filter. Notably, what factors influence the likelihood of hTMEM175 adopting open or closed states is an open question. Due to hTMEM175's association with lysosomal homeostasis and the development of Parkinson's Disease (*Blauwendraat et al., 2019*; *Cang et al., 2015*; *Iwaki et al., 2019*; *Jinn et al., 2019*; *Jinn et al., 2017*; *Nalls et al., 2014*), future studies will be necessary to identify stimuli that regulate gating and to understand how these stimuli alter the equilibrium between open and closed states to regulate lysosomal $K^+$ flux.

# Materials and methods

**Key resources table**

| Reagent type (species) or resource | Designation | Source or reference | Identifiers | Additional information |
|---|---|---|---|---|
| Gene (*Homo sapiens*) | *hTMEM175* | *Synbio technologies* | | |
| Cell line (*H. sapiens*) | HEK-293T | ATCC | CRL-3216 RRID:CVCL_0063 | |
| Cell line (*H. sapiens*) | HEK-293S GnTi- | ATCC | CRL-3022 | |
| Chemical compound, drug | 1-palmitoyl-2-oleoyl-sn-glycero-3-phosphoethanolamine | Avanti Polar Lipids | 850757 | |

*Continued on next page*

*Continued*

| Reagent type (species) or resource | Designation | Source or reference | Identifiers | Additional information |
|---|---|---|---|---|
| Chemical compound, drug | 1-palmitoyl-2-oleoyl-sn-glycero-3-phospho-(1'-rac-glycerol) (sodium salt) | Avanti Polar Lipids | 840457 | |
| Chemical compound, drug | carbonyl cyanide m-chlorophenylhydrazone (CCCP) | Thermo Fisher Scientific | 215911250 MG | |
| Chemical compound, drug | 9-amino-6-chloro-2-methoxyacridine (ACMA) | Thermo Fisher Scientific | A1324 | |
| Chemical compound, drug | valinomycin | Sigma | V0627 | |
| Chemical compound, drug | Polyethylenimine, Linear, MW 25000, Transfection Grade (PEI 25K) | Polysciences, Inc | 23966–1 | |
| Chemical compound, drug | Sodium Butyrate | Sigma | 8451440100 | |
| Chemical compound, drug | lauryl maltoside neopentyl glycol | Anatrace | NG310 | |
| Chemical compound, drug | n-Octyl-β-D-Maltopyranoside | Anatrace | O310S | |
| Software, algorithm | MotionCor2 | *Zheng et al., 2017* | RRID:SCR_016499 | |
| Software, algorithm | CtfFind 4.1.10 | *Rohou and Grigorieff, 2015* | RRID:SCR_016731 | |
| Software, algorithm | RELION 3.1 | *Scheres, 2016* | http://www2.mrc-lmb.cam.ac.uk/relion RRID:SCR_016274 | |
| Software, algorithm | SerialEM | *Mastronarde, 2005* | RRID:SCR_017293 | |
| Software, algorithm | cryoSPARC v2 | Structura Biotechnology | https://cryosparc.com/ RRID:SCR_016501 | |
| Software, algorithm | PHENIX | *Liebschner et al., 2019* | https://www.phenix-online.org/ RRID:SCR_014224 | |
| Software, algorithm | COOT | *Emsley et al., 2010* | https://www2.mrc-lmb.cam.ac.uk/personal/pemsley/coot/ RRID:SCR_014222 | |
| Software, algorithm | PyMOL | *Schrödinger, 2020* | https://pymol.org/2/ RRID:SCR_000305 | |
| Software, algorithm | UCSF Chimera | *Pettersen et al., 2004* | https://www.cgl.ucsf.edu/chimera RRID:SCR_004097 | |
| Software, algorithm | GraphPad Prism 7 | GraphPad Software | | |
| Software, algorithm | SoftMax Pro 6 | Molecular Devices | | |
| Software, algorithm | Axon Digidata 1550B digitizer | Molecular Devices | | |
| Software, algorithm | Clampex 10.6 | Molecular Devices | | |
| Software, algorithm | CHAP | *Klesse et al., 2019* | https://www.channotation.org/ | |
| Software, algorithm | Clampfit 10.6 | Molecular Devices | | |
| Others | QUANTIFOIL R1.2/1.3 holey carbon grids | Quantifoil | | |
| Others | FEI Vitrobot Mark IV | FEI Thermo Fisher | | |

## Protein expression and purification

The gene encoding human TMEM175 was synthesized (SynBio) and subcloned into a BacMam expression vector with a C-terminal EGFP-tag fused via a short linker containing a PreScission protease site (*Goehring et al., 2014*). The plasmid was mixed with PEI 25 k (Polysciences, Inc) for 30 min and then used to transfect HEK293S GnTi⁻ cells (ATCC CRL-3022). After 24 hr incubation at 37 ˚C, sodium butyrate was added to a final concentration of 10 mM, and cells were allowed to grow at 37 ˚C for an additional 48–72 hr before harvesting. Cell pellets were washed in phosphate-buffered saline solution and flash frozen in liquid nitrogen. Expressed protein was solubilized in 2% lauryl maltose neopentyl glycol (LMNG, Anatrace), 20 mM HEPES pH 7.5, 150 mM KCl supplemented with protease-inhibitor cocktail (1 mM PMSF, 2.5 µg/mL aprotinin, 2.5 µg/mL leupeptin, 1 µg/mL pepstatin A) and DNase. Solubilized protein was separated by centrifugation 74,766 *g* for 40 mins, followed by binding to anti-GFP nanobody resin for 2 hr. Anti-GFP nanobody affinity chromatography was performed by 20 column volumes of washing with buffer containing 0.1% LMNG, 20 mM HEPES pH 7.5, 150 mM KCl, 2 mM DTT, followed by overnight PreScission digestion, and elution with wash buffer. Eluted protein was further purified by size exclusion chromatography on a Superdex 200 Increase 10/300 GL (GE healthcare) in SEC buffer (0.1% LMNG, 50 mM Tris pH 8.0, 150 mM KCl, 2 mM DTT). Peak fractions were pooled and concentrated to ~4 mg/mL using CORNING SPIN-X concentrators (100 kDa cutoff). For the CsCl samples, KCl was replaced with CsCl for all steps of the purification.

## Proteoliposome reconstitution and flux assay

1-palmitoyl-2-oleoyl-sn-glycero-3-phosphoethanolamine (POPE) and 1-palmitoyl-2-oleoyl-sn-glycero-3-phospho-(1'-rac-glycerol) (POPG) in chloroform (Avanti) were mixed in a ratio 3:1 (mg:mg) and dried under argon gas. The dried lipid mixture was solubilized in pentane and dried again under argon gas to remove residual chloroform. Dried lipids were then desiccated for 2 hr under vacuum. Lipids were resuspended in 10 mM Hepes pH 7.4, 300 mM KCl to a final concentration of 10 mg/ml. Unilamellar vesicles were formed by sonication and then solubilized using 8% (w/v) octyl maltoside. Full length hTMEM175 purified in LMNG at a concentration of 1 mg/ml was mixed with the octyl maltoside-solubilized lipids and dialyzed using 25 kDa MWC bags (SpectraPor) in 10 mM Hepes pH 7.4, 300 mM KCl, 2 mM dithiothreitol (DTT) for 5 days with daily exchange of dialysis buffer. After dialysis, harvested proteoliposomes were snap frozen in liquid nitrogen and stored at −80 ˚C until use. Proteoliposomes were rapidly thawed at 37 ˚C, sonicated for 5 s, incubated at room temperature for 2–4 hr before use, and then diluted 100-fold into a flux assay buffer composed of 10 mM Hepes pH 7.4, 300 mM NaCl, 0.2 µM 9-amino-6-chloro-2-methoxyacridine (ACMA).

Data were collected on a SpectraMax M5 fluorometer (Molecular Devices) using Softmax Pro six software. ACMA excitation/emission wavelengths were 410/490 nm, respectively. Fluorescence intensity measurements were collected every 30 s. The ionophore CCCP (1 µM) and valinomycin (20 nM) were added at 150 s and 600 s, respectively.

## Electron microscopy sample preparation and data acquisition

4–5 µl of purified hTMEM175 at a concentration of 4 mg/ml was applied to glow-discharged Au 400 mesh QUANTIFOIL R1.2/1.3 holey carbon grids (Quantifoil, and then plunged into liquid nitrogen-cooled liquid ethane with an FEI Vitrobot Mark IV (FEI Thermo Fisher). Grids were transferred to a 300 keV FEI Titan Krios microscopy equipped with a K2 summit direct electron detector (Gatan). Images were recorded with SerialEM (*Mastronarde, 2005*) in super-resolution mode at 22,500x, corresponding to pixel size of 0.544 Å. Dose rate was eight electrons/pixel/s, and defocus range was 1.2–2.5 µm. Images were recorded for 8 s with 0.2 s subframes (total 40 subframes), corresponding to a total dose of 61 electrons/Å$^2$.

## Electron microscopy data processing

40-frame super-resolution movies (0.544 Å/pixel) of TMEM175 in KCl were gain corrected, Fourier cropped by two and aligned using whole-frame and local motion correction algorithms by Motioncor2 (*Zheng et al., 2017*) (1.088 Å/pixel). Whole-frame CTF parameters were determined using CTFfind 4.1.10 (*Rohou and Grigorieff, 2015*). Approximately 500 particles were manually selected to generate initial templates for autopicking that were improved by several rounds of two-

dimensional classification in Relion 3.0 (*Scheres, 2016*), resulting in 2,499,425 particles for KCl data set 1 and 1,654,189 particles for KCl data set 2. False-positive selections and contaminants were excluded from the data using multiple rounds of heterogeneous classification in cryoSPARC v2 (*Punjani et al., 2017*) using models generated from the ab initio algorithm in cryoSPARC v2, resulting in a stack of 571,468 particles. After particle polishing in Relion and local CTF estimation and higher order aberration correction in cryoSPARC v2, a consensus reconstruction was determined at resolution of 2.7 Å. 3D variability analysis in cryoSPARC v2 was then employed to characterize conformational heterogeneity, revealing two states that were subsequently used for iterative rounds of supervised heterogeneous refinement in cryoSPARC v2. The final stack for class 1 contained 342,340 particles and yielded a reconstruction with an estimated resolution of 2.6 Å by non-uniform refinement in cryoSPARC v2 (*Punjani et al., 2019*). The final stack for class 2 contained 70,132 particles and yielded a reconstruction with an estimated resolution of 3.0 Å by non-uniform refinement in cryoSPARC v2. The final reconstructions of class 1 and class 2 were further improved by employing density modification on the two unfiltered half-maps with a soft mask in Phenix (*Terwilliger et al., 2019*).

40-frame super-resolution movies (0.544 Å/pixel) of TMEM175 in CsCl were gain corrected, Fourier cropped by two and aligned using whole-frame and local motion correction algorithms by Motioncor2 (1.088 Å/pixel). Approximately 500 particles were manually selected to generate initial templates for autopicking that were improved by several rounds of two-dimensional classification in Relion and autopicking using Relion, resulting in 2,537,436 particles for CsCl data set 1 and 1,737,783 particles for CsCl data set 2. False-positive selections and contaminants were excluded through iterative rounds of heterogeneous classification in cryoSPARC v2 using models generated from the ab initio algorithm in cryoSPARC v2, resulting in a stack of 330,698 particles. After particle polishing in Relion and local CTF estimation and higher order aberration correction in cryoSPARC v2, a consensus reconstruction was determined to 3.1 Å. 3D variability analysis in cryoSPARC v2 was then employed to characterize conformational heterogeneity, revealing two states that were used for iterative rounds of seeded heterogeneous refinement in cryoSPARC v2. The final stack for class 1 contained 104,126 particles and yielded a reconstruction with an estimated resolution of 3.2 Å by non-uniform refinement in cryoSPARC v2. The final stack for class 2 contained 70,132 particles and yielded a reconstruction with an estimated resolution of 3.2 Å by non-uniform refinement in cryoSPARC v2. The final reconstructions of class 1 and class 2 were further improved by employing density modification on the two unfiltered half-maps with a soft mask that includes the detergent micelle in Phenix.

## Model building and coordinate refinement

Poly-alanine helices were manually built into the transmembrane helices of the class 1 $K^+$ density map using coot (*Emsley et al., 2010*). The helices were manually registered using large side chains and the connecting loops were manually built into the density. Densities corresponding to TM5 and TM6 (residues 174–251) were too poorly ordered and omitted from the model. The final model contains residues 30–173, and 254–476. Four ions were assigned by identifying overlapping non-protein density peaks in the class 1 $K^+$ and $Cs^+$ maps. Atomic coordinates were refined against the density modified map using phenix.real_space_refinement with geometric and Ramachandran restraints maintained throughout (*Adams et al., 2010*).

The refined class 1 structure was manually docked into the class 2 density map using chimera (*Pettersen et al., 2004*). The model was manually rebuilt using coot to fit the density. Three ions were assigned by identifying overlapping non-protein density peaks in the $K^+$ and $Cs^+$ maps. Water molecules were placed into the remaining non-protein density peaks. Atomic coordinates were refined against the density modified map using phenix.real_space_refinement with geometric and Ramachandran restraints maintained throughout (*Adams et al., 2010*).

The $Cs^+$ class 1 and class 2 structures were determined by docking in the $K^+$ class 1 and class 2 structure in Coot and manually rebuilding the protein to best fit the density map. Atomic coordinates were refined against the density modified map using phenix.real_space_refinement with geometric and Ramachandran restraints maintained throughout (*Adams et al., 2010*).

## Fluorescence size exclusion chromatography (FSEC)

Plasmids encoding GFP-tagged versions of wild-type and mutant hTMEM175 were mixed with PEI 25 k (Polysciences, Inc) for 30 min and then used to transfect HEK293S GnTi⁻ cells. After 24 hr incubation at 37 °C, sodium butyrate was added to a final concentration of 10 mM, and cells were allowed to grow at 37 °C for an additional 48–72 hr before harvesting. Cell pellets were washed in phosphate-buffered saline solution and flash frozen in liquid nitrogen. Expressed protein was solubilized in 2% lauryl maltose neopentyl glycol (LMNG), 20 mM HEPES pH 7.5, 150 mM KCl supplemented with protease-inhibitor cocktail (1 mM PMSF, 2.5 μg/mL aprotinin, 2.5 μg/mL leupeptin, 1 μg/mL pepstatin A) and DNase. Solubilized protein was separated by centrifugation 21,130 $g$ for 60 mins. Separated proteins were injected to and monitored by fluorescence size exclusion chromatography on a Superose 6 Increase 10/300 GL (GE healthcare) in SEC buffer (0.1% LMNG, 50 mM Tris pH 8.0, 150 mM KCl, 2 mM DTT). Fluorescence was monitored at 488/509 nm of excitation/emission wavelength, respectively.

## Electrophysiology

Electrophysiological recordings of hTMEM175 constructs were performed in HEK293T cells (ATCC CRL-3216). HEK293T cells cultured in DMEM supplemented with 10% FBS were transfected with 2 μg of hTMEM175 plasmid using 6 μg of PEI 25 k (Polysciences, Inc). 24–48 hr following transfection, cells were detached by trypsin treatment. The detached cells were transferred to poly-Lys-treated 35 mm single dishes (FluoroDish, World Precision Instruments) and incubated overnight at 37 °C in fresh media. Immediately prior to recording, media was replaced with a bath solution containing 145 mM Na- methanesulfonate (MS), 5 mM NaCl or KCl, 1 mM MgCl₂, 1 mM CaCl₂, 10 mM HEPES/Tris pH 7.4. 10 cm long borosilicate glass were pulled and fire polished (Sutter instrument). The resistance of glass pipette was 5 ~ 8 MΩ were filled with a pipette solution containing 150 mM K-MS or Cs-MS, 5 mM MgCl₂, 10 mM EGTA/Tris, 10 mM HEPES/Tris pH 7.4, GΩ seals were formed after gentle suction. The recordings were performed in whole cell patch clamp configuration using the following protocol: from a holding potential of 0 mV, the voltage was stepped to voltages between −100 and +100 mV, in 20 mV increments. The currents were recorded using Axon Digidata 1550B digitizer and Clampex 10.6 (Molecular Devices, LLC) and analyzed using Clampfit 10.6 (Molecular Devices, LLC). Each experiment was performed a unique cell and currents were normalized to the maximum current of each experiment (at +100 mV). Each condition includes cells from at least two independent transfections.

In bi-ionic conditions, the relative permeability between cations are calculated using the following equations.

$$\frac{P_X}{P_Y} = \frac{[Y^+]_{ext}}{[X^+]_{int}} e^{-\left(\frac{E_{rev}F}{RT}\right)}$$

Where $P_X$ and $P_Y$ are the permeabilities of intracellular and extracellular cation X and Y, respectively, $E_{rev}$ is the measured reversal potential, F is Faradays' constant, R is the gas constant, and T is the absolute temperature.

Figures were prepared with UCSF Chimera (*Pettersen et al., 2004*), PyMol (Schrödinger), CHAP (*Klesse et al., 2019*) and Prism 7 (GraphPad).

## Acknowledgements

We thank M de la Cruz at The MSKCC Richard Rifkind Center for cryo-EM for assistance with data collection, the MSKCC HPC group for assistance with data processing and SB Long and T Walz for comments on the manuscript. This work was supported in part by NIH-NCI Cancer Center Support Grant (P30 CA008748), the Josie Robertson Investigators Program (to RKH) and the Searle Scholars Program (to RKH).

## Additional information

### Funding

| Funder | Grant reference number | Author |
|---|---|---|
| Searle Scholars Program | | Richard K Hite |
| Memorial Sloan Kettering Cancer Center | Josie Robertson Investigators Program | Richard K Hite |
| National Cancer Institute | P30 CA008748 | Richard K Hite |

The funders had no role in study design, data collection and interpretation, or the decision to submit the work for publication.

### Author contributions

SeCheol Oh, Conceptualization, Data curation, Formal analysis, Validation, Investigation, Methodology; Navid Paknejad, Data curation, Methodology; Richard K Hite, Conceptualization, Formal analysis, Supervision, Funding acquisition, Investigation, Methodology, Project administration

### Author ORCIDs

SeCheol Oh (iD) https://orcid.org/0000-0002-1685-5922
Richard K Hite (iD) https://orcid.org/0000-0003-0496-0669

### Decision letter and Author response

Decision letter https://doi.org/10.7554/eLife.53430.sa1
Author response https://doi.org/10.7554/eLife.53430.sa2

## Additional files

### Supplementary files

- Transparent reporting form

### Data availability

Cryo-EM maps and atomic coordinates have been deposited with the EMDB and PDB under accession codes EMDB-21603 and PDB 6WC9 for Class 1 TMEM175 in $K^+$, codes EMDB-21604 and PDB 6WCA for Class 2 TMEM175 in $K^+$, codes EMDB-21605 and PDB 6WCB for Class 1 TMEM175 in $Cs^+$ and codes EMDB-21606 and PDB 6WCC for Class 2 TMEM175 in $Cs^+$. All other reagents are available from the corresponding author upon reasonable request.

The following datasets were generated:

| Author(s) | Year | Dataset title | Dataset URL | Database and Identifier |
|---|---|---|---|---|
| Oh SC, Paknejad N, Hite RK | 2020 | Cryo-EM structure of human TMEM175 in an open state in $K^+$ | http://www.ebi.ac.uk/pdbe/entry/emdb/EMD-21603 | Electron Microscopy Data Bank, EMDB-21603 |
| Oh SC, Paknejad N, Hite RK | 2020 | Cryo-EM structure of human TMEM175 in an closed state in $K^+$ | http://www.ebi.ac.uk/pdbe/entry/emdb/EMD-21604 | Electron Microscopy Data Bank, EMDB-21604 |
| Oh SC, Paknejad N, Hite RK | 2020 | Cryo-EM structure of human TMEM175 in an open state in $K^+$ | http://www.rcsb.org/structure/6WC9 | RCSB Protein Data Bank, 6WC9 |
| Oh SC, Paknejad N, Hite RK | 2020 | Cryo-EM structure of human TMEM175 in an closed state in $K^+$ | http://www.rcsb.org/structure/6WCA | RCSB Protein Data Bank, 6WCA |
| Oh SC, Paknejad N, Hite RK | 2020 | Cryo-EM structure of human TMEM175 in an open state in $Cs^+$ | http://www.ebi.ac.uk/pdbe/entry/emdb/EMD-21605 | Electron Microscopy Data Bank, EMDB-21605 |
| Oh SC, Paknejad N, Hite RK | 2020 | Cryo-EM structure of human TMEM175 in an closed state in $Cs^+$ | http://www.ebi.ac.uk/pdbe/entry/emdb/EMD- | Electron Microscopy Data Bank, EMDB- |

| | | | | 21606 | 21606 |
|---|---|---|---|---|---|
| Oh SC, Paknejad N, Hite RK | 2020 | Cryo-EM structure of human TMEM175 in an open state in Cs$^+$ | | http://www.rcsb.org/structure/6WCB | RCSB Protein Data Bank, 6WCB |
| Oh SC, Paknejad N, Hite RK | 2020 | Cryo-EM structure of human TMEM175 in an closed state in Cs$^+$ | | http://www.rcsb.org/structure/6WCC | RCSB Protein Data Bank, 6WCC |

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
