## [Decision Letter]

**Acceptance summary:**

Although the human lysosomal potassium ion channel, TMEM175, lacks a canonical selectivity filter, it is 40-fold more selective to potassium than sodium. In a canonical K^+^ selectivity filter, K^+^ ions bind preferentially to the ion binding sites in the filter. This study reveals a potentially alternate mechanism of ion selectivity wherein the cost of desolvation of an ion as it enters a hydrophobic constriction is the most important determinant. K^+^ can be desolvated more easily than Na^+^ and therefore these channels are potassium selective.

**Decision letter after peer review:**

Thank you for submitting your article "Structural basis for permeation and selectivity in the non-canonical human lysosomal K^+^ channel TMEM175" for consideration by *eLife*. Your article has been reviewed by Richard Aldrich as the Senior Editor, a Reviewing Editor, and three reviewers.

The reviewers have discussed the reviews with one another and the Reviewing Editor has drafted this decision to help you prepare a revised submission.

Summary:

This study represents a significant contribution to our understanding of ion selectivity in a recently discovered new family of K^+^-permeable channels, the TMEM175 potassium channel family, which is present in all three kingdoms of life. In prokaryotes, these channels display modest potassium over sodium selectivity but in humans the TMEM175 is ~ 40-folds more selective for K^+^ than for Na^+^ ions. Interestingly, these novel channels lack the typical signature sequence (i.e., TTVGYGD) that is characteristic of the highly K^+^ selective channels, which are exemplified by the shaker and KcsA channels. Although structures of the prokaryotic channels have been solved recently, the ion-binding sites have not been observed in those structures. The authors solved structures of this channel in several conditions: (1) high potassium, (2) low pH, and (3) high Cs. The high K^+^ structure is at a resolution below 3 Å, and very high quality. There are several interesting aspects to this structure. First, two of the transmembrane segments from one of the two 6-TM repeats in the dimer are disordered and cannot be modeled, while the corresponding two segments from the other repeat are very well ordered. Second, despite the fact that the pore appears very narrow with the narrowest constriction formed by 4 isoleucines where only dehydrated potassium can squeeze by, there are many non-protein densities in the pore that the authors attribute to water and K^+^. To distinguish between them, they determined the structure in Cs^+^, under the assumption that since Cs^+^ scatters electrons stronger than K^+^, hTMEM175 in Cs^+^ should display strong peaks at the Cs^+^ binding sites, which should be in the same locations as K^+^, since Cs^+^ permeates the channel as well. They identified 5 K^+^ binding sites, where 3 are on the symmetry axis and 2 are off it but none of them are fully coordinated by protein atoms. They thus conclude that the selectivity arises from the transient dehydration of the ions as they pass through the narrowest Ile constriction. Overall, this study contributes new knowledge that will ultimately lead to a broader understanding of the mechanisms of ion selectivity. Nevertheless, all the reviewers have expressed concerns that the interpretation of the data is some instances is too narrow and a more nuanced discussion is in order.

Essential revisions:

1) One of the key points here is the role of TMEM175 "hydrophobic constrictions" as sieving barriers for discriminating between the various dehydrated cations. This is difficult to reconcile with our existing knowledge of ion permeation even though a similar mechanism has been proposed previously by Jiang and his colleagues. One possibility is these channels are in a closed conformation and that these hydrophobic constrictions act as dewetted gates (as opposed to steric gates). Although mutations in the hydrophobic residues alter ion conduction, it is not clear whether these mutations modify the structure. I would like the authors to rationalize their point of view and discuss it in the context of dewetting and the possibility that the hydrophobic constriction could be hydrophobic gates (Aryal et al., 2015).

2) Related to the above point. Please add a paragraph in the Discussion section acknowledging the possibility that the channel is in a closed state in the structures presented and that there may be additional determinants for selectivity in this channel family.

3) In subsection “Human TMEM175 is highly selective for K^+^”, the ACMA assay to test the function of the TMEM175 was done in POPE:POPG liposomes and the phospholipid composition so the lysosome is dramatically different. Can the authors argue why did they do the assay in this lipid composition? Why not to perform the ACMA assay as it has been done before for the Hv1 channel? The Mackinnon group performed the assay in a lipid composition that mimic the membrane of human neutrophil plasma membrane, which is more physiologically relevant.

4) Subsection “Structure of the ion conduction pathway”, the authors concluded too strong that because the T274V was nonconductive, "the water-mediated ion binding site, which is disrupted in the T274V mutant is essential for ion permeation". Although it is indeed very provocative to state this conclusion, we can imagine several possible scenarios in which the T274V halt ion conduction beside disrupting the water-mediated ion binding site. I advise the authors to be more conservative and less conclusive since in the absence of the T274V structure, they cannot categorically conclude this. Other mutations could have been done (as T274S), and the cryo-EM structures of these mutants could have been solved to support the mechanism proposed by the authors.

5) The authors attempted to use a low resolution cryo-EM structure soaked in Cesium to assign the K^+^-binding sites in the TMTM175 structure. This experimental approach assume that the putative 5 ion binding sites are equivalents and that they have the same ion selectivity. It has been shown that in KcsA, the second binding site although highly selective for potassium ions, at steady state conditions, does not coordinate Cesium or Rubidium ions. So, the authors should explain to the reader the limitation(s) of this experimental approach since it could draw incorrect conclusions. For example, the Cs4 does not directly correspond to densities found on the hTMEM175k density map and that could be explain by a different type of interaction between Cs^+^ ions and the protein at that specific site.

6) In Subsection “Structure of the ion conduction pathway” the authors concluded that because the cryo-EM structure at pH 5.5, that mimics a late endosomes and lysosomes, seems to be identical to the hTMEM175k (obtained at basic pH), then these results are consistent with a channel no gated by pH. I agree to disagree here, since there are many assumptions and generalization in this sentence. For starters, nobody knows how the hTMEM175 channel is gated. The very simplistic and superficial functional studies of the hTMEM175 solely showed non-inactivating and non-voltage dependent macroscopic currents. But these whole cell currents could arise for channels that are inactivated but at the steady state have a sizable rate of recovery from the inactive state. One could argue that the absence of a lipid bilayer precluded the structural changes elicited by the acidic pH. The lack of structural changes does not categorically prove the point of Oh et al. Hence, I would advise a more conservative narrative that include other reasons of why the structures at different pHs are identical. Finally, it is very intriguing why the protonation state of the acidic residues that coordinate K1 (i.e., D279 or E282), which are very close to the luminal side, did not change the occupancy of the ion at the K1 site. I would argue that in the absence of a lipid bilayer, the local concentration of protons is significantly lower in the cryo-EM structure and therefore the ion occupancy of the K1 site is the same as the one obtained at basic pH.

7) The authors proposed that negative charges on the channel help to select cations over anions. However, there is no experimental data to support this claim. Does the selectivity change if these charges are neutralized? If this model is true, the channel ion selectivity should change as a function of the pHs, as the protonation state of these residues is changed.

8) I would like the authors to discuss the fact that the TMEM175 normally works with a deltapH, which is acidic at the luminal side (pH of ~4.8). How does this fact affect their conclusions about the lack of pH-dependent putative structural changes? In physiologically relevant conditions the channel confront to dramatically different environments in regard to the proton concentration.

9) I am concerned about the reversal potential measurements. If the seals have some degree of leak, the authors could be wrong estimating the reversal potential. For instance, if the cells have some chloride leak, as usually happen, your reversal potential is going to be altered. A leak subtraction protocol (P over N) could solve this problem. As the ion selectivity measurements are an important dataset in the present work, I think the authors should make sure that the reported measurements are fully reliable, hence a very clear explanation how they were performed must be provided.

10) Lack of functional activity in a mutant channel, such as the T274V or the I46M/I271M mutants can arise due to an effect on channel folding/assembly in the membrane. An interpretation of the mutants requires that an effect of the amino acid substitution on folding/assembly is ruled out.

11) A I46N/I274N substitution has been shown to result in a loss of ionic selectivity (in Lee, 2017). This result should be described and discussed in the manuscript.

[Editors' note: further revisions were suggested prior to acceptance, as described below.]

Thank you for resubmitting your work entitled "Gating and selectivity mechanisms for the lysosomal K^+^ channel TMEM175" for further consideration by *eLife*. Your revised article has been evaluated by Richard Aldrich (Senior Editor) and a Reviewing Editor.

The manuscript has been improved but there are some remaining issues that need to be addressed before acceptance, as outlined below:

1) The reviewers agree that the manuscript has improved significantly, and the additional data may provide insight into the mechanism of gating in TMEM175. Nonetheless, the reviewers remain concerned that you are over interpreting your data and want you to tone down the claims. While your discovery is certainly exciting, it is important to convey to a general reader the uncertainties associated with interpretations based primarily on the structure. Given that there is no single channel data or additional biochemistry to provide a "functional" framework for interpreting this structural data, it is necessary that you explicitly address these caveats.

2) Even with regards to ion selectivity, one has to be careful because the open structure may not correspond to an open state unless it is validated in another way. If that is the case, then the mechanism of ion selectivity as proposed may not stand. Therefore, please make sure this caveat is taken into consideration.

3) The figure numbers are out registry i.e., in subsection “Ion-conduction pathway contains ordered ions and waters” it reads.…"Glu259 via ordered water (Figure 4A)" it should be (Figure 5A).

4) Can the authors add the distance between the ions on Figure 4A, B and D?

5) The assignment of the density at the Ile constriction as K^+^ is highly speculative. The density is very weak in the Cs^+^ map and is very poor when the C2 symmetry is not applied. It is also hard to imagine that K^+^ ions would be stable in an hydrophobic environment. The authors need to mention that the assignment of the density as K^+^ is speculative.

6) The FSEC profiles shown in Figure 3—figure supplement 1A are saturating the detector (flat at the top). These curves cannot be used to claim that the mutants were expressed at similar levels to wild type as the levels can be different but not seen due to detector saturation. That claim should be dropped. The curves do indicate that the mutants fold similar to the wild type.

---

## [Author Response]

Summary:This study represents a significant contribution to our understanding of ion selectivity in a recently discovered new family of K^+^-permeable channels, the TMEM175 potassium channel family, which is present in all three kingdoms of life. In prokaryotes, these channels display modest potassium over sodium selectivity but in humans the TMEM175 is ~ 40-folds more selective for K^+^ than for Na^+^ ions. Interestingly, these novel channels lack the typical signature sequence (i.e., TTVGYGD) that is characteristic of the highly K^+^ selective channels, which are exemplified by the shaker and KcsA channels. Although structures of the prokaryotic channels have been solved recently, the ion-binding sites have not been observed in those structures. The authors solved structures of this channel in several conditions: (1) high potassium, (2) low pH, and (3) high Cs. The high K^+^ structure is at a resolution below 3 Å, and very high quality. There are several interesting aspects to this structure. First, two of the transmembrane segments from one of the two 6-TM repeats in the dimer are disordered and cannot be modeled, while the corresponding two segments from the other repeat are very well ordered. Second, despite the fact that the pore appears very narrow with the narrowest constriction formed by 4 isoleucines where only dehydrated potassium can squeeze by, there are many non-protein densities in the pore that the authors attribute to water and K^+^. To distinguish between them, they determined the structure in Cs, under the assumption that since Cs^+^ scatters electrons stronger than K^+^, hTMEM175 in Cs^+^ should display strong peaks at the Cs binding sites, which should be in the same locations as K^+^, since Cs^+^ permeates the channel as well. They identified 5 K^+^ binding sites, where 3 are on the symmetry axis and 2 are off it but none of them are fully coordinated by protein atoms. They thus conclude that the selectivity arises from the transient dehydration of the ions as they pass through the narrowest Ile constriction. Overall, this study contributes new knowledge that will ultimately lead to a broader understanding of the mechanisms of ion selectivity. Nevertheless, all the reviewers have expressed concerns that the interpretation of the data is some instances is too narrow and a more nuanced discussion is in order.

We thank the editors and reviewers for their positive assessment of our manuscript and their detailed and constructive comments. To address some of the reviewers’ concerns, we developed a new image processing workflow to better extract structural information from our cryo-EM images. The re-processing improved the yield of “good” particles and enabled us to resolve a novel conformational state. This new conformation, which we call class 2, exists in the both the KCl and CsCl data sets. By removing the class 2 particles from the KCl data set, we were able to obtain a more homogenous particle population. Refinement of these particles resulted in a 2.6 Å reconstruction, which we call class 1 and is similar to the structure that we described in our initial submission. Comparison of the class 1 and class 2 states reveals significant conformational changes in the pore, including changes to Ile46 and Ile271. In class 2, Ile46 and Ile271 form a constriction whose minimum radius is 0.5 Å that blocks ion permeation through the pore. In class 1, the isoleucine constriction widens to a minimum radius of 1.6 Å and density corresponding to a K^+^ ion is resolved within the constriction. The ion density is elongated, extending throughout the constriction, consistent with the ion occupying multiple binding sites. Ordered water molecules are resolved on either side of the constriction that can coordinate the ion as it enters the constriction from either side. We propose that class 1 represents a conductive state and that the rigid architecture of isoleucine constriction necessitates the transient dehydration of ions, thereby favoring K^+^ over Na^+^. Thus, our work represents a major advance over previous studies of TMEM175 homologs by answering two of the major outstanding questions regarding human TMEM175 function – how it is gated and how it can selectively permeate potassium ions.

Due to the substantial changes in the data, we have extensively rewritten the manuscript and modified several of figures. Our point-by-point responses to the reviewers’ comments are listed below.

Essential revisions:1) One of the key points here is the role of TMEM175 "hydrophobic constrictions" as sieving barriers for discriminating between the various dehydrated cations. This is difficult to reconcile with our existing knowledge of ion permeation even though a similar mechanism has been proposed previously by Jiang and his colleagues. One possibility is these channels are in a closed conformation and that these hydrophobic constrictions act as dewetted gates (as opposed to steric gates). Although mutations in the hydrophobic residues alter ion conduction, it is not clear whether these mutations modify the structure. I would like the authors to rationalize their point of view and discuss it in the context of dewetting and the possibility that the hydrophobic constriction could be hydrophobic gates (Aryal et al., 2015).

We thank the reviewers for their careful consideration of our proposed selectivity mechanism. To characterize the role of the constriction formed by Ile46 and Ile271 in ion selectivity and channel gating, we re-processed our data and identified a novel conformation, which we call class 2. In this new class 2 state, Ile46 and Ile271 constrict the pore to a minimum radius of 0.5 Å, which is too narrow to accommodate cations, even in a dehydrated state (Figure 5). We therefore conclude that class 2 represents a closed state and that Ile46 and Ile271 form a steric gate to ion permeation.

In the higher-resolution class 1 density map that we describe in the revised manuscript, the isoleucine constriction adopts a wider conformation than in class 2 and density corresponding to a K^+^ ion is resolved (Figure 3, Figure 4 and Figure 5). The K^+^ ion density is elongated, rather than spherical, consistent with the ion partially occupying multiple different positions. Density is also resolved for ordered water molecules on either side of the constriction that are positioned to coordinate ions as they approach the constriction. Using whole-cell patch clamp, we assessed the function of channels possessing mutations to several of the residues that coordinate the water molecules near the isoleucine constriction. In both cases, mutations that disrupt the ability of these residues to participate in the coordination of these waters (S45A and T274V) result in non-functional channels, while conservative mutations (S45T and T274S) yield functional channels (Figure 6B). We therefore propose that class 1 represents a conductive state and that Ile46 and Ile271 form a constriction that selectively permeates cations based on their relative enthalpies of dehydration.

Together, these structures reveal two roles for the isoleucine constriction. In the closed structure, the constriction forms a steric gate that prevents ion permeation. In the open structure, the constriction widens are forms a selectivity filter.

2) Related to the above point. Please add a paragraph in the Discussion section acknowledging the possibility that the channel is in a closed state in the structures presented and that there may be additional determinants for selectivity in this channel family.

In the revised version of the manuscript, we describe structures of hTMEM175 in two conformations. In the closed class 2 state, the channel is gated by a closure of the isoleucine constriction (Figure 5C). In the class 1 state, ion density is resolved within the widened isoleucine constriction (Figure 3D). The ion in the constriction is coordinated by water molecules that reside on either side of the constriction that can coordinate ions as the enter the constriction from either side (Figure 6). In the revised manuscript, we discuss in subsection “Structural heterogeneity reveals gating mechanism” these features of the structure and propose that class 1 represents a conductive state of hTMEM175.

In addition to the isoleucine constriction, our electrophysiological analysis of wild-type and mutant TMEM175 channels is consistent with Thr274 also participating in ion selectivity. Mutation of Thr274 to serine right-shifted the reversal potential from ~-65 mV for wild-type hTMEM175 to ~-34 mV.

3) In subsection “Human TMEM175 is highly selective for K^+^”, the ACMA assay to test the function of the TMEM175 was done in POPE:POPG liposomes and the phospholipid composition so the lysosome is dramatically different. Can the authors argue why did they do the assay in this lipid composition? Why not to perform the ACMA assay as it has been done before for the Hv1 channel? The Mackinnon group performed the assay in a lipid composition that mimic the membrane of human neutrophil plasma membrane, which is more physiologically relevant.

We agree with the reviewers that POPE:POPG liposomes do not reflect the physiological composition of the lysosomal membrane. Instead, we performed the proteolipsome flux assay using channels reconstituted into POPE:POPG liposomes, as has been used for analysis of a number of other ion channels. For the proteolipsome flux assay that we are using, POPE:POPG liposomes are particularly advantageous because they do not allow proton leak. In contrast, protons can readily permeate through protein-free liposomes of several commonly used lipid mixtures, including *E. coli* polar lipids and those containing PC lipids. Because PC is a major constituent of endolysosomal membranes (Escribá et al., 2015), which would generate background leak in our assay, we performed our experiments with POPE:POPG liposomes. In future studies, it will be essential to characterize the influence of different lipids on the function of hTMEM175.

4) Subsection “Structure of the ion conduction pathway”, the authors concluded too strong that because the T274V was nonconductive, "the water-mediated ion binding site, which is disrupted in the T274V mutant is essential for ion permeation". Although it is indeed very provocative to state this conclusion, we can imagine several possible scenarios in which the T274V halt ion conduction beside disrupting the water-mediated ion binding site. I advise the authors to be more conservative and less conclusive since in the absence of the T274V structure, they cannot categorically conclude this. Other mutations could have been done (as T274S), and the cryo-EM structures of these mutants could have been solved to support the mechanism proposed by the authors.

To further investigate the role of water molecules in the permeation of hTMEM175, we performed additional mutagenesis experiments. In addition to our analysis of the T274V mutant, we examined a more conservative T274S mutant as suggested by the reviewers as well as conservative and non-conservative mutations to Ser45, whose side chain participates in the coordination of water molecules on the cytosolic side of the isoleucine constriction. We first compared the folding of these mutants to wild-type hTMEM175 by FSEC analysis (Goehring et al., 2014), which revealed that all of the mutants eluted single peaks with retention volumes similar to wild type (Figure S8). We next examined their ability to selectively permeate Cs^+^ ions using whole-cell patch clamp. Cs^+^-selective currents could be detected from cells expressing the conservative S45T or T274S mutations while cells expressing the non-conservative S45A or T274V mutants displayed current levels that were indistinguishable from background currents measured in non-transfected cells (Figure 6B-C). While it remains possible that the S45A and T274V mutations alter trafficking to the plasma membrane, it is unlikely as the conservative mutants are both localized to the plasma membrane. Thus, we interpret the lack of exogenous currents from cells expressing the S45A and T274V mutants as resulting from defects in channel function.

5) The authors attempted to use a low resolution cryo-EM structure soaked in Cesium to assign the K^+^-binding sites in the TMTM175 structure. This experimental approach assume that the putative 5 ion binding sites are equivalents and that they have the same ion selectivity. It has been shown that in KcsA, the second binding site although highly selective for potassium ions, at steady state conditions, does not coordinate Cesium or Rubidium ions. So, the authors should explain to the reader the limitation(s) of this experimental approach since it could draw incorrect conclusions. For example, the Cs4 does not directly correspond to densities found on the hTMEM175k density map and that could be explain by a different type of interaction between Cs^+^ ions and the protein at that specific site.

We agree with the reviewers that the Cs^+^ is imperfect mimic of K^+^ and that it may bind to proteins differently. This is a fundamental limitation of our approach and we have expanded on this limitation in the Discussion section.

6) In Subsection “Structure of the ion conduction pathway” the authors concluded that because the cryo-EM structure at pH 5.5, that mimics a late endosomes and lysosomes, seems to be identical to the hTMEM175k (obtained at basic pH), then these results are consistent with a channel no gated by pH. I agree to disagree here, since there are many assumptions and generalization in this sentence. For starters, nobody knows how the hTMEM175 channel is gated. The very simplistic and superficial functional studies of the hTMEM175 solely showed non-inactivating and non-voltage dependent macroscopic currents. But these whole cell currents could arise for channels that are inactivated but at the steady state have a sizable rate of recovery from the inactive state. One could argue that the absence of a lipid bilayer precluded the structural changes elicited by the acidic pH. The lack of structural changes does not categorically prove the point of Oh et al. Hence, I would advise a more conservative narrative that include other reasons of why the structures at different pHs are identical. Finally, it is very intriguing why the protonation state of the acidic residues that coordinate K1 (i.e., D279 or E282), which are very close to the luminal side, did not change the occupancy of the ion at the K1 site. I would argue that in the absence of a lipid bilayer, the local concentration of protons is significantly lower in the cryo-EM structure and therefore the ion occupancy of the K1 site is the same as the one obtained at basic pH.

We agree with the reviewers that the gating of hTMEM175 has yet to be functionally well characterized and it remains unclear how pH will alter channel activity. Our functional analyses of hTMEM175 were performed at pH 7.4 and thus do not inform on channel activity at pH 5.5. Because the focus of our manuscript has changed with the addition of the closed structure and because we have not performed a detailed functional characterization of hTMEM175 in acidic conditions, we have excluded the structure of hTMEM175 in the pH 5.5 condition, which is nearly identical to the pH 8.0 condition from our manuscript.

7) The authors proposed that negative charges on the channel help to select cations over anions. However, there is no experimental data to support this claim. Does the selectivity change if these charges are neutralized? If this model is true, the channel ion selectivity should change as a function of the pHs, as the protonation state of these residues is changed.

We agree with the reviewers that our proposed mechanism for cation over anion selectivity was speculative and have removed this discussion from the manuscript.

8) I would like the authors to discuss the fact that the TMEM175 normally works with a deltapH, which is acidic at the luminal side (pH of ~4.8). How does this fact affect their conclusions about the lack of pH-dependent putative structural changes? In physiologically relevant conditions the channel confront to dramatically different environments in regard to the proton concentration.

As mentioned for point 6, we have removed the pH 5.5 structure from the manuscript. We would therefore prefer not to speculate on the effects of a deltapH on channel gating.

9) I am concerned about the reversal potential measurements. If the seals have some degree of leak, the authors could be wrong estimating the reversal potential. For instance, if the cells have some chloride leak, as usually happen, your reversal potential is going to be altered. A leak subtraction protocol (P over N) could solve this problem. As the ion selectivity measurements are an important dataset in the present work, I think the authors should make sure that the reported measurements are fully reliable, hence a very clear explanation how they were performed must be provided.

We agree with the reviewers that systematic errors exist in our reversal potential measurements due to endogenous background currents. Our experiments were conducted using bath solution containing 145 mM Na-methanesulfonate (MS) 5 mM NaCl or KCl, 1 mM MgCl_2_, 1 mM CaCl_2_, 10 mM HEPES/Tris pH 7.4. and a pipette solution containing 150 mM KMS or Cs-MS, 5 mM MgCl_2_, 10 mM EGTA/Tris, 10 mM HEPES/Tris pH 7.4. The magnitude of the chloride component of these currents is reduced by using methanesulfonate as the predominant anion.

As suggested by the reviewers, we attempted to perform a subtraction experiment using the mean background current of the non-transfected cells. However, as we show in Figure 2—figure supplement 1, the magnitude of the background currents of the non-transfected cells varied substantially and the subtractions resulted in traces with clear artefacts and were not interpretable. Because these systematic errors in our permeability measurement could not be corrected using a background subtraction protocol, we emphasize in subsection “Human TMEM175 is highly selective for K^+^” in the manuscript that these measurements underestimate the true reversal potential.

10) Lack of functional activity in a mutant channel, such as the T274V or the I46M/I271M mutants can arise due to an effect on channel folding/assembly in the membrane. An interpretation of the mutants requires that an effect of the amino acid substitution on folding/assembly is ruled out.

As described in our response to point 4, we used FSEC to analyze the hydrodynamic radius of GFP-tagged mutant hTMEM175 channels that we characterized by electrophysiology. In all cases, the channels eluted as a single peak with a similar retention volume to wild-type hTMEM175. We interpret this result as the mutants all retaining their dimer architecture (Figure 4—figure supplement 1).

11) A I46N/I274N substitution has been shown to result in a loss of ionic selectivity (in Lee, 2017). This result should be described and discussed in the manuscript.

We agree with the reviewers and have included the I46N/I274N mutant in our discussion of ion selectivity of TMEM175 channels in subsection “Ion binding sites in the pore”.

[Editors' note: further revisions were suggested prior to acceptance, as described below.]

1) The reviewers agree that the manuscript has improved significantly, and the additional data may provide insight into the mechanism of gating in TMEM175. Nonetheless, the reviewers remain concerned that you are over interpreting your data and want you to tone down the claims. While your discovery is certainly exciting, it is important to convey to a general reader the uncertainties associated with interpretations based primarily on the structure. Given that there is no single channel data or additional biochemistry to provide a "functional" framework for interpreting this structural data, it is necessary that you explicitly address these caveats.

We thank the reviewers for the critical assessment of our work. We have added a paragraph to the discussion addressing some of the caveats of structural characterizations of our work on TMEM175 and of ion channels in general.

2) Even with regards to ion selectivity, one has to be careful because the open structure may not correspond to an open state unless it is validated in another way. If that is the case, then the mechanism of ion selectivity as proposed may not stand. Therefore, please make sure this caveat is taken into consideration.

We agree with the reviewers that due to the inherent difficulties in assigning discrete functional states to static cryo-EM structures, it is not possible to be certain that our open structure may correspond to an open state. We have revised the discussion to better address the possibility that our functional assignments are speculative.

3) The figure numbers are out registry i.e., in subsection “Ion-conduction pathway contains ordered ions and waters” it reads.…"Glu 259 via ordered water (Figure 4A)" it should be (Figure 5A).

Thank you very much, we have corrected the figure citations.

4) Can the authors add the distance between the ions on Figure 4A, B and D?

Thank you very much for this suggestion, we have added the distances between the ions in the figures. Furthermore, because the distances between symmetry-related K1, K2 and K4 ion binding sites are ~3 Å, we have noted in the text at the end of the third paragraph in subsection “Ion-conduction pathway contains ordered ions and waters” that it is unlikely that both of the symmetry-related ion binding sites are simultaneously occupied.

5) The assignment of the density at the Ile constriction as K^+^ is highly speculative. The density is very weak in the Cs^+^ map and is very poor when the C2 symmetry is not applied. It is also hard to imagine that K^+^ ions would be stable in a hydrophobic environment. The authors need to mention that the assignment of the density as K^+^ is speculative.

We thank the reviewers for their comment. We agree that the ion binding site near the Ile constriction is the weakest of the four sites resolved in both density maps and that it is unusual ion binding site being so closely positioned to the hydrophobic Ile side chains. We have revised the text in subsection “Ion-conduction pathway contains ordered ions and waters” to highlight that ions will only transiently access the hydrophobic environment of the isoleucine constriction. The ions will more stably bind at the cytoplasmic or luminal sides of the constriction where they can be partially coordinated by ordered waters.

6) The FSEC profiles shown in Figure 3—figure supplement 1A are saturating the detector (flat at the top). These curves cannot be used to claim that the mutants were expressed at similar levels to wild type as the levels can be different but not seen due to detector saturation. That claim should be dropped. The curves do indicate that the mutants fold similar to the wild type.

We have repeated the FSEC analysis using a lower laser gain power in which the detector is not saturated. The results demonstrate that with the exception of T274V, the mutant constructs express at similar levels as wild-type TMEM175. We have modified the text in subsection “Ion-conduction pathway contains ordered ions and waters” and replaced figure 3—figure supplement 1A with the updated panel.